# $\mu$LO: Compute-Efficient Meta-Generalization of Learned Optimizers

**Benjamin Thérien**[1,2]     **Charles-Étienne Joseph**[1,2]     **Boris Knyazev**[1,2,4]
**Edouard Oyallon**[5]     **Irina Rish**[1,2]     **Eugene Belilovsky**[2,3]

[1]Université de Montréal; [2]Mila – Quebec AI Institute; [3]Concordia University, Montréal;
[4]Samsung AI Lab, Montréal; [5]ISIR, Sorbonne University, CNRS, Paris, France.

## Abstract

Learned optimizers (LOs) have the potential to significantly reduce the wall-clock training time of neural networks. However, they can struggle to optimize unseen tasks (*meta-generalize*), especially when training networks wider than those seen during meta-training. To address this, we derive the Maximal Update Parametrization ($\mu$P) for two state-of-the-art learned optimizer architectures and propose a simple meta-training recipe for $\mu$-parameterized LOs ($\mu$LOs). Our empirical evaluation demonstrates that LOs meta-trained with our recipe substantially improve meta-generalization to wider unseen tasks when compared to LOs trained under standard parametrization (SP) using the same compute budget. We also empirically observe that $\mu$LOs exhibit unexpectedly improved meta-generalization to deeper networks ($5\times$ meta-training) and surprising generalization to much longer training horizons ($25\times$ meta-training) when compared to SP LOs.

## 1 Introduction

While deep learning (DL) has largely replaced hand-designed algorithms, one crucial component of DL training remains hand-crafted: gradient-based optimizers. While popular optimizers such as Adam or SGD provably converge to a local minimum in non-convex settings (Kingma & Ba, 2017; Li et al., 2023a; Robbins, 1951), the existing literature provides no evidence that these optimizers converge to the global optimum at the optimal rate. With the lack of theory certifying the optimality of existing optimizers and the clear strength of data-driven methods, it is natural to turn towards data-driven solutions for improving the optimization of neural networks.

Taking a step in this direction, Andrychowicz et al. (2016); Wichrowska et al. (2017); Metz et al. (2019; 2022a) replace hand-designed optimizers with small neural networks called learned optimizers (LOs). LOs are meta-learned on a task distribution by minimizing the loss of the inner learning problem (e.g. neural network training in our case) across a batch of tasks. Being neural networks themselves, these optimizers are advantaged by their substantially larger parameter counts than Adam or SGD, making them suitable to large-scale meta-training. For instance, Metz et al. (2022b) showed that scaling up learned optimizer meta-training to 4000 TPU months can produce an optimizer, VeLO, that significantly outperforms well-tuned hand-designed optimizers without requiring hyperparameter tuning. However, even VeLO has limitations in *meta-generalization* – optimizing unseen problems. Specifically, VeLO (Metz et al., 2022b) is known to (1) have difficulty optimizing models much wider and deeper than those seen during meta-training (See Figures 6 and 9 of Metz et al. (2022b)) and (2) generalize poorly to longer optimization problems (e.g., training for more steps) than those seen during meta-training.

The problem of **meta-generalization** is fundamental to learned optimization due to the requirement for tractable meta-training and the expectation of strong performance across a combinatorially large set of downstream tasks. Meta-generalization refers to the ability of a meta-learned algorithm to *generalize*, that is, perform well when applied to unseen tasks. In the case of LOs, a learned optimizer

---

Correspondence to: Benjamin Thérien ⟨benjamin.therien@umontreal.ca⟩ and Eugene Belilovsky ⟨eugene.belilovsky@concordia.ca⟩. Our code is open-sourced: https://github.com/bentherien/mu_learned_optimization.

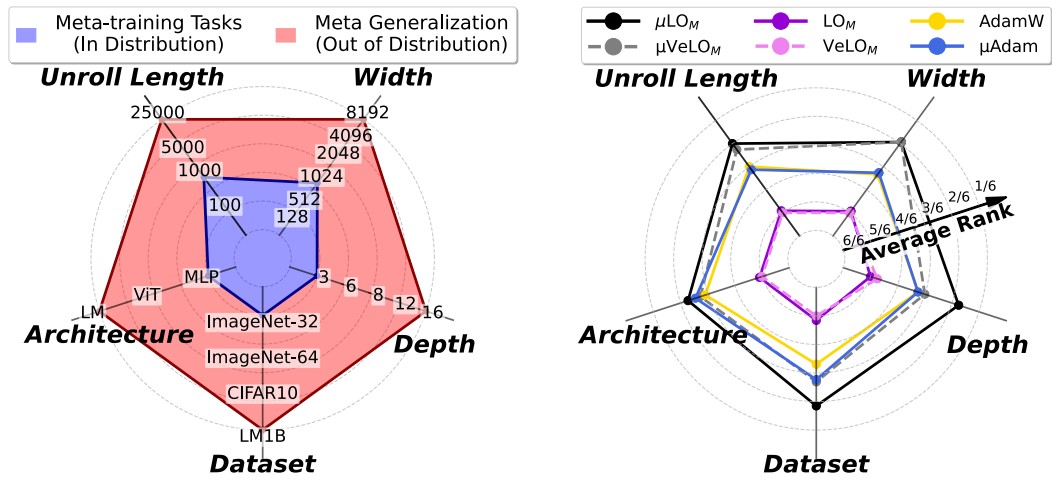

(a) Axes of Meta-Generalization                  (b) Performance by Average Rank

Figure 1: **Meta-generalization is severely limited without our approach.** Subfigure (a) illustrates *meta-generalization* axes by distinguishing between meta-training tasks used herein (blue) and out-of-distribution tasks (red). Subfigure (b) reports the average rank across tasks within our evaluation suite that are out-of-distribution with respect to the corresponding axis. Both AdamW and $\mu$Adam undergo task-specific hyperparameter tuning across more than $500$ configurations per task. Learned Optimizers of the same architecture are meta-learned on the same tasks with a FLOP-matched budget.

trained on a tractable and, thus, limited distribution of meta-training tasks should nevertheless exhibit strong performance when applied to out-of-distribution tasks: new combinations of architecture, dataset, and training objective (Figure 1). Even changes as small as increasing the hidden dimension of the architecture (width), the number of layers (depth), or the number of training steps (unroll length) can cause meaningful distribution shifts between meta-training and testing tasks, leading to poor generalization. Consequently, understanding and improving meta-generalization is central to making learned optimizers practical for real-world machine learning workloads.

In this work, we focus on the problem of LO meta-generalization to tasks of larger hidden dimension (width) than those seen during meta-learning. A related problem is that of transferring hyperparameters of hand-designed optimizers to wider tasks. Introduced by Yang et al. (2022), $\mu$P is an optimizer-dependent and width-dependent parameterization (e.g., a rule for initializing a model, scaling its pre-activations, and scaling the optimizer's updates) that allows hyperparameter transfer to larger width tasks for Adam and SGD. Making the connection between hyperparameter-transfer and meta-generalization, we ask: *Are existing learned optimizer architectures compatible with $\mu$P? Does meta-learning optimizers under $\mu$P improve meta-generalization?* To answer this question, we theoretically analyze two recent LO architectures (Metz et al., 2022a;b) (sec. 4), derive the appropriate maximal update parameterization for them, and carefully design a low-cost meta-training recipe to bring out their meta-generalization capabilities. We then provide extensive experimental evidence demonstrating that $\mu$LOs generalize to large unseen tasks. Our contributions are as follows:

- We derive $\mu$-parameterization for two popular learned optimizer architectures (VeLO and small_fc_lopt) and demonstrate theoretically that our parameterization satisfies $\mu$P desiderata.

- We design a set of meta-training and meta-testing tasks enabling a systematic study of meta-generalization and demonstrate that our $\mu$LOs significantly outperform strong baseline LOs and hand-designed optimizers.

- We demonstrate empirically that our $\mu$LOs show surprisingly good generalization to deeper networks ($5\times$ meta-training) and longer training horizons ($25\times$ meta-training) when compared to baseline LOs.

## 2 BACKGROUND

**Learned optimizer objective.** A standard approach to learning optimizers (Metz et al., 2019) is to solve the following meta-learning problem:

$$\min_\phi \; \mathbb{E}_{(\mathcal{D},\mathcal{L},\boldsymbol{w}_0)\sim\mathcal{T}} \left[ \mathbb{E}_{(X,Y)\sim\mathcal{D}} \left[ \frac{1}{T} \sum_{t=0}^{T-1} \mathcal{L}(X,Y; f_\phi(\boldsymbol{u}_t), \boldsymbol{w}_t) \right] \right]. \tag{1}$$

Where $\mathcal{T}$ is a distribution over optimization tasks defined as tuples of dataset $\mathcal{D}$, objective function $\mathcal{L}$, and initial weights $\boldsymbol{w}_0$ associated with a particular neural architecture (we refer to this network as the *optimizee*); $\phi$ represents the weights of the learned optimizer, $f_\phi$ with input features $\boldsymbol{u}_t$; and $T$ is the length of the unroll which we write as a fixed quantity for simplicity. In equation 1 and in our experiments, the sum of per-timestep loss is the quantity being optimized. It should be noted, however, that one could also optimize the final loss, final accuracy, or any other performance metric. Gradient descent is the preferred approach to solving equation 1. However, estimating meta-gradients via backpropagation is known to be problematic for long unrolls (Metz et al., 2019). Therefore, learned optimizer meta-gradients are estimated using evolution strategies and their variants (Vicol et al., 2021; Buckman et al., 2018; Nesterov & Spokoiny, 2017; Parmas et al., 2018; Vicol, 2023; Li et al., 2023b).

**Learned optimizer input, output, and update.** Learned optimizer neural architectures have taken many forms over the years, we will briefly review two recent architectures, **small_fc_lopt** of Metz et al. (2022a) and **VeLO** of Metz et al. (2022b), as they are used in our experiments. These learned optimizers construct input features $\boldsymbol{u}_t$ based on momentum accumulators, a variance accumulator, and multiple adafactor accumulators, we provide a full list in Tables 2, 3, and 4 of the Appendix. At every gradient descent step, small_fc_lopt and VeLO are applied to each parameter of the optimizee, producing two outputs: the magnitude ($m$) and direction ($d$) of the update. VeLO additionally outputs a tensor-level learning rate, $\alpha_{\boldsymbol{W}}$. The per-parameter update for both optimizers is given as

$$w_t = w_{t-1} - \alpha_{\boldsymbol{W}} \lambda_1 d \exp(\lambda_2 m), \tag{2}$$

where $w$ is a parameter of weight matrix $\boldsymbol{W}$, $\lambda_1$ and $\lambda_2$ are constant values set to $0.001$ to bias initial step sizes to be small. For small_fc_lopt, $\alpha_{\boldsymbol{W}} = 1$ always. We refer interested readers to appendix sections A.1.1 and A.1.2 for more details.

## 3 RELATED WORK

**Generalization in LOs.** There are three main difficulties of learned optimizer generalization (Chen et al., 2022; Amos, 2022): (1) optimizing unseen tasks; (2) optimizing beyond maximum unroll length seen during meta-training; (3) training optimizees that do not overfit. Among these, (3) has been most extensively addressed as this problem has been well studied in classic optimization literature. For example, extra-regularization terms can be directly applied to a learned optimizer (Harrison et al., 2022; Yang et al., 2023). In addition, (3) can be addressed by meta-training on a validation set objective (Metz et al., 2019) or parameterizing LOs as hyperparameter controllers (Almeida et al., 2021). The problem (2) has been mitigated by regularization (Harrison et al., 2022; Yang et al., 2023) and larger-scale meta-training (Metz et al., 2022b). However, (1) has remained a more difficult and understudied problem.

To the best of our knowledge, the only current approach to tackle this problem is to meta-train LOs on thousands of tasks (Metz et al., 2022b). However, this approach is extremely expensive and seems bound to fail in the regime where the optimizer is expected to generalize from small meta-training tasks in standard parameterization to large unseen tasks: figures 6 and 9 of Metz et al. (2022b) demonstrate that this was not achieved even when using 4000 TPU-months of compute. Generalization would be expected if all tasks, no matter the size, were included in the meta-training distribution, but such an approach is simply intractable and is likely to remain so.

**Maximal Update Parametrization and Hyperparameter transfer.** First proposed by Yang & Hu (2021), the Maximal Update Parametrization is the unique stable abc-Parametrization where every layer learns features. The parameterization was derived for adaptive optimizers by Yang & Littwin (2023) and was applied by Yang et al. (2022) to enable zero-shot hyperparameter transfer

for Adam and SGD. Most recently, in tensor programs VI, Yang et al. (2024) propose Depth-$\mu$P, a parameterization allowing for hyperparameter transfer in infinitely deep networks. While it is appealing, Depth-$\mu$P is only valid for residual networks with a block depth of 1, so it does not apply most practical architectures (e.g., transformers, resnets, etc.). For these reasons, we do not study Depth-$\mu$P herein. Following from the original discovery of hyperparameter transfer in Yang et al. (2022), a number of follow-up works have emerged that are not part of the tensor programs series. Dey et al. (2024) investigates transferring hyperparameters across different sparsity levels and widths. Blake et al. (2025) investigates a combination of $\mu$P and unit scaling, which results in easier tuning and more stable low-precision training. Everett et al. (2024) investigate the alignment assumptions of Yang et al. (2022) and find that appropriate per-layer learning rate prescriptions can also enable hyperparameter transfer in standard, mean field, and NTK parameterizations. In their empirical investigation of scaling exponents across these parameterizations, the authors find that SP with layer-wise learning rates outperforms $\mu$P. While we study the impact of meta-learning optimizers in $\mu$P on meta-generalization herein, it is still an open question which parameterization is best for meta-learning optimizers. Finally, in concurrent work, Dey et al. (2025) propose CompleteP, a parameterization that can achieve transfer of optimal hyperparameters across depth and width.

## 4    $\mu$-PARAMETRIZATION FOR LEARNED OPTIMIZERS

Parameterizing an optimizee neural network in $\mu$P requires special handling of the initialization variance, pre-activation multipliers, and optimizer update for each weight matrix $\boldsymbol{W} \in \mathbb{R}^{n \times m}$ in the network. Specifically, these quantities will depend on the functional form of the optimizer and the dependence of $n$ (FAN_OUT) and $m$ (FAN_IN) on width. We will refer to weight matrices in a network of width $h$ as hidden layers if $\Theta(n) = \Theta(m) = \Theta(h)$, as output layers if $\Theta(n) = \Theta(1) \wedge \Theta(m) = \Theta(h)$, and as input layers if $\Theta(n) = \Theta(h) \wedge \Theta(m) = \Theta(1)$. Here, $\Theta$ is standard asymptotic notation. Note that all biases and the weights of normalization layers are considered input layers and should be scaled as such. With this in mind, consider an arbitrary neural network[1] whose weight matrices are denoted $\boldsymbol{W}_l$, where $l$ indexes the layers; the following modifications are then required to obtain $\mu$P for learned optimizers.

**Optimizee Initialization-$\mu$.**  If $\boldsymbol{W}_l$ belongs to a hidden or input layer, its weights should be initialized as $\mathcal{N}(0, \frac{1}{\text{FAN\_IN}})$. Output layers should have their weights initialized as $\mathcal{N}(0, 1)$.

**Optimizee Multipliers-$\mu$.**   Output layer pre-activations should be multiplied by $\frac{1}{\text{FAN\_IN}}$ during the forward pass.

**Optimizer Update Scaling-$\mu$.**   The learned optimizer's update (eq. 2) is re-scaled as follows:

$$w_t = \begin{cases} w_{t-1} - \frac{1}{\text{FAN\_IN}} \cdot \left( \alpha_{\boldsymbol{W}_l} \lambda_1 d \exp\left(\lambda_2 m\right) \right) & \boldsymbol{W}_l \text{ is a hidden layer} \\ w_{t-1} - \alpha_{\boldsymbol{W}_l} \lambda_1 d \exp\left(\lambda_2 m\right) & \text{otherwise.} \end{cases} \tag{3}$$

Where $w$ is a parameter of the weight matrix, $\boldsymbol{W}_l$, and the dependence of $d$ and $m$ on $w_{t-1}$ is not made explicit for simplicity. For transfer to the largest width optimizees, it may also become necessary to re-scale numerical underflow constants ($\epsilon$) by $\frac{1}{\text{FAN\_IN}}$ as suggested by (Everett et al., 2024). However, for the scales reported on by our experiments, we did not find this to be necessary.

We now prove that our parameterization satisfies the $\mu$P Desiderata ((Yang et al., 2022) Sec. J.2.1).

**Proposition 4.1** ($small\_fc\_lopt$ $\mu$P). *Assume that the Learned Optimizer $f_\phi$ has the form $small\_fc\_lopt$ is fed with features given in Appendix A.1.1 and that during training the optimizee's parameters and input data become aligned, leading to Law of Large Numbers (LLN) scaling, then the update, initialization, and pre-activation multiplier above is sufficient to obtain a Maximal Update Parametrization.*

**Proposition 4.2** (VeLO $\mu$P). *Assume that $\phi$ in Proposition 4.1 is generated using an LSTM with the input features described in Appendix A.1.2 and that during training the optimizee's parameters and input data become aligned, leading to Law of Large Numbers (LLN) scaling, then the update, initialization, and pre-activation multiplier above is sufficient to obtain a Maximal Update Parametrization.*

*Proof.* The proof is provided in Appendix A.2.    □

---

[1] The $\mu$LO parameterization can be applied to any neural network architecture.

# 5 EMPIRICAL EVALUATION

We construct a suite of optimization tasks of varying width to evaluate the meta-generalization properties of our $\mu$LOs meta-trained on MLPs vs per-task tuned $\mu$Adam (Yang et al., 2022), per-task tuned SP AdamW (Loshchilov & Hutter, 2019), and baseline SP LOs (meta-trained on MLP tasks). Our main focus is to evaluate meta-generalization to wider networks as this is a key weakness of learned optimizers in previous works. However, we also establish the generalization properties of $\mu$LOs to deeper networks and longer training horizons. Please note that while $\mu$LOs inherit the theoretical properties of $\mu$P for width scaling, our findings with respect to deeper networks and longer training are purely empirical.

## 5.1 SETUP

**Baseline LOs and $\mu$LOs.** The meta-training configuration of each learned optimizer is summarized in Table 5. Each learned optimizer (ours and the baselines) in our empirical evaluation is meta-trained using the multiple-width single-task meta-training recipe proposed in section 5.2.1. **Notably, these tasks only include MLPs (see Fig 1), while the hand-desinged optimizers in our study are tuned individually on each task.** The SP baselines sheds light on whether simply varying the SP optimizee width during meta-training is enough to achieve generalization of the LO to wider networks in SP. During meta-training, we set the inner problem length to be 1000 iterations. Therefore, any optimization beyond this length is considered out-of-distribution. For all meta-training and hyperparameter tuning details, including ablation experiments, see section C of the appendix.

$\mu$**Adam** is a strong hand-designed $\mu$P baseline. It follows the Adam $\mu$-parametrization and does not use weight decay as this is incompatible with $\mu$P (Yang et al., 2022). $\mu$Adam is tuned on a width=1024 version of each task as this is the width of the largest meta-training task seen by our learned optimizers (see Table 5). We tune the learning rate ($\eta$) and accumulator coefficients ($\beta_1$ and $\beta_2$) using a grid search over more than 500 different configurations. This is repeated once for each task in our suite. Section B.1 of the appendix provides more details about the grid search including the values swept and the best values found.

**AdamW** (Loshchilov & Hutter, 2019) is a strong hand-designed SP baseline. It is tuned on the largest meta-training task seen by our learned optimizers (Table 5). AdamW is tuned on a width=1024 version of each task as this is the width of the largest meta-training task seen by our learned optimizers (see Table 5). We tune the learning rate ($\eta$), accumulator coefficients ($\beta_1$ and $\beta_2$), and weight decay ($\lambda$) using a grid search over more than 500 different configurations. This is repeated once for each task in our suite. Section B.2 of the appendix provides more details about the grid search including the values swept and the best values found.

**Evaluation tasks.** Our evaluation suite includes 35 tasks spanning image classification (CIFAR-10, ImageNet) using MLPs and Vision Transformers (ViTs) (Dosovitskiy et al., 2020) and autoregressive language modeling with a decoder-only transformer on LM1B (Chelba et al., 2013). To create the tasks, we further vary image size (for image classification), width, and depth of the optimizee network, and the number of optimization steps. See Table 10 of the appendix for an extended description of all the tasks.

## 5.2 RESULTS

In the following sections, we evaluate different meta-training distributions for training $\mu$LOs (Sec. 5.2.1); we present results empirically verifying the pre-activation stability of our $\mu$LOs (Sec. 5.2.2); we present the results of our main empirical evaluation of meta-generalization to wider networks (Sec. 5.2.2); a study of $\mu$LOs generalization to deeper networks (Sec. 5.2.4); and a study of $\mu$LOs generalization to longer training horizons (Sec. 5.2.4). All of our figures report training loss and report the average loss across 5 random seeds. Each seed corresponds to a different ordering of training data and a different initialization of the optimizee. All error bars in our plots report standard error across seeds. Standard error is $\frac{\sigma}{\sqrt{n}}$ where $\sigma$ is the population standard deviation and $n$ is the number of samples.

### 5.2.1 EVALUATING META-TRAINING DISTRIBUTIONS FOR $\mu$LOS

In $\mu$-transfer (Yang et al., 2022), hyperparameters are typically tuned on a small proxy task before being transferred to the large target task. In contrast, learned optimizers are typically meta-trained

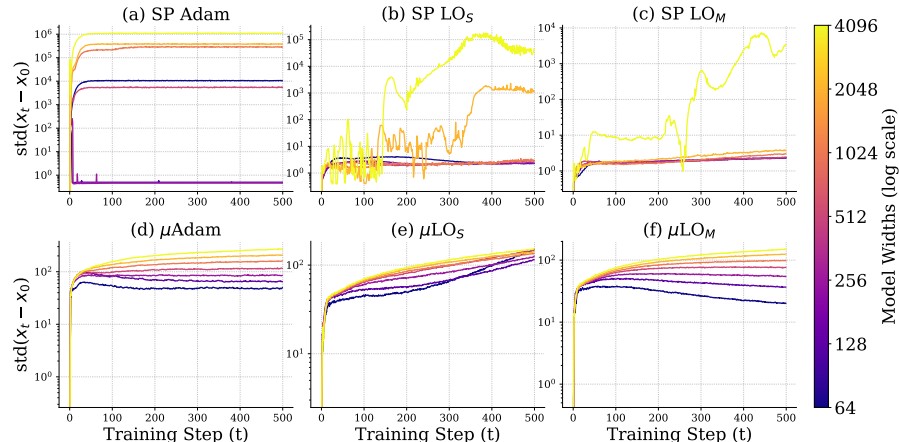

Figure 2: **Layer 2 pre-activations behave harmoniously in $\mu$P for $\mu$LOs and $\mu$Adam alike.** We report the evolution of coordinate-wise standard deviation of the difference between the initial ($t = 0$) and $t$-th second-layer pre-activations of an MLP during training for the first 500 steps of a single run (the remaining layers behave similarly, see Sec. G). We observe that all models parameterized in $\mu$P enjoy stable coordinates across widths, while the pre-activations of larger-width models in SP blow up after a number of training steps.

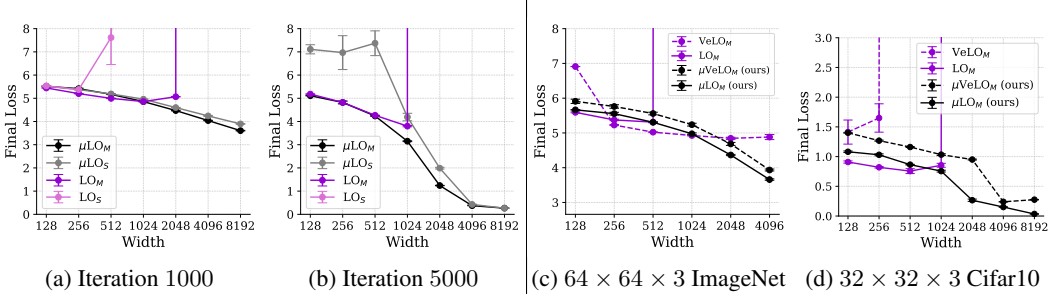

Figure 3: **Generalization beyond meta-training widths is severely limited without our approach.** Each point is the average final training loss over 5 seeds with standard error bars. Subfigures (a) and (b) report the results of our meta-training task ablation on the ImageNet-32 meta-training tasks at 1000 and 5000 steps. Subfigures (c) and (d) report the performance of $\mu$LO$_M$ and $\mu$VeLO$_M$ on OOD datasets.

on a distribution of tasks. To verify the effectiveness of each approach for meta-training $\mu$LOs, we compare $\mu$LO$_S$, meta-trained on a single width=128 MLP ImageNet classification task (see Tab. 5), to $\mu$LO$_M$, meta-trained on width $\in \{128, 512, 1024\}$ MLP ImageNet classification tasks. Each optimizer targets 1000 step problems. We include equivalent standard parameterization baselines for reference (LO$_S$ and LO$_M$). Figure 3 reports the performance of each optimizer on a suite of MLP classification tasks of increasing width. When training for 1000 steps (meta-training unroll length), we observe that $\mu$LO$_M$ outperforms $\mu$LO$_S$ as the width of the model is increased (Fig. 3 (a)). Moreover, we observe that there is a discrepancy in performance between both models after 5000 steps (Fig. 3 (b)), showing that meta-training with multiple tasks of different widths has benefits for generalization to longer unrolls in addition to improved generalization to larger optimizees. Given the improved generalization of $\mu$LO$_M$ compared to $\mu$LO$_S$, we adopt the multiple-width meta-training recipe as part of our method. Subsequent experiments (e.g., Figures 3 and 4) will show that our recipe is also effective for meta-training $\mu$VeLO.

### 5.2.2 EVALUATING PRE-ACTIVATION STABILITY

We now verify that desiderata J.1 of Yang et al. (2022) is satisfied empirically. In Figure 2, we report the evolution of the coordinate-wise standard deviation of the difference between initial (t=0) and current (t) second-layer pre-activations of an MLP during the first 500 steps of training for a

single trial. We observe that all models parameterized in $\mu$P enjoy stable coordinates across widths, suggesting that desiderata J.1 is satisfied by our parameterization. In contrast, the pre-activations of the larger MLPs in SP blow up immediately for SP Adam while they take noticeably longer for $LO_S$ and $LO_M$. Section G of the appendix contains similar plots for the remaining layers of the MLP which show similar trends. In summary, we find, empirically, that pre-activations of $\mu$LOs and $\mu$Adam are similarly stable across widths, while the activations of SP Adam and SP LOs both blow up but behave qualitatively differently.

### 5.2.3 Meta-generalization to wider networks

Given our goal of improving LO generalization to unseen wider tasks, the bulk of our empirical evaluation is presented in this section. Specifically, we evaluate the behavior of $\mu$LOs as the width of tasks increases well beyond what was seen during meta-training. To accomplish this, we fix the depth of each task and vary the width (see Table 10 for a full list of tasks), leading to a testbed of 32 different tasks. We then train each task using the baselines and $\mu$-optimizers outlined in section 5 for 5000 steps for 5 different random seeds. This involves training 1120 different neural networks. To make the results easily digestible, we summarize them by width and final performance in Figure 4 and by average optimizer rank in Table 1. We also highlight the smooth training dynamics of our optimizers at the largest widths in Figure 4.

**Performance measured by final loss as a function of width.** Figure 3 compares the training loss after 1000 steps of SP learned optimizers to $\mu$-parameterized learned optimizers for different widths. This is shown in three subfigures for three MLP image classification tasks: (a) Imagenet $32 \times 32 \times 3$ (IN32), (c) Imagenet $64 \times 64 \times 3$ (IN64), and (d) Cifar-10 $32 \times 32 \times 3$ (C10). Subfigure (a) shows the performance of learned optimizers on larger versions of the meta-training tasks. We observe that the $\mu$LOs achieve lower final training loss as the width of the task is increased. In contrast, $LO_M$ diverges for widths larger than 2048. Subfigure (b) evaluates our $\mu$LOs on $64 \times 64 \times 3$ ImageNet images (e.g., when the input width is larger). Similarly, we observe smooth improvements in the loss as the optimizee width increases for $\mu$LOs, while their SP counterparts either diverge at width 512 ($LO_M$) or fail to substantially improve the loss beyond width 1024 ($VeLO_M$). Finally, Subfigure (c) shows the performance of our $\mu$LOs on Cifar-10 (smaller output width) as the width is increased. Similarly, we observe smooth improvements in the loss as the width increases for $\mu$LOs, while their SP counterparts either diverge immediately at small widths ($VeLO_M$) or diverge by width 1024 ($LO_M$).

**Training dynamics at the largest widths** Figure 4 reports the training curves of different optimizers on the largest width tasks in our suite. Despite training for $5\times$ longer than the maximum meta-training unroll length, our $\mu$LOs are capable of smoothly decreasing the loss for the largest out-of-distribution tasks in our suite. In contrast, the strong SP LO baselines diverge by 1000 steps (subfigures (a),(b),(c),(d)), or fail to decrease the training loss (subfigure (e)), demonstrating the clear benefit of $\mu$LOs for learned optimization. Our $\mu$LOs also substantially best the per-task-tuned AdamW and $\mu$Adam baselines (subfigures (a) and (b)), match the best performing hand-designed optimizer in subfigure (c), and nearly matches or outperforms the strongest hand-designed baseline performance on far out-of-distribution LM and ViT tasks (subfigures (d) and (e)). These results demonstrate that, under our $\mu$LO meta-training recipe, learning optimizers that smoothly train large neural networks (e.g., demonstrated an 8B parameter model typically uses width=4096) is possible at low cost ($\mu LO_M$ is meta-trained for 100 GPU hours).

Table 1: **Summary of optimizer performance on large tasks.** We report the average rank of different optimizers across the five tasks in our suite. We evaluate each optimizer on large-width tasks: Large (2048), XL (4096 for MLPs and 3072 for vit and LM), and XXL (largest size for each task see Tab. 10 of the appendix). We bold the strongest, underline the second strongest, and italicize the third strongest average rank in each column. We observe that, across all iterations, $\mu LO_M$ and $\mu VeLO_M$ consistently obtain the best and second-best ranks for all tasks.

| Optimizer | Loss at 1k steps | | | Loss at 3k steps | | | Loss at 5k steps | | |
| | OoD (Large) | OoD (XL) | OoD (XXL) | OoD (Large) | OoD (XL) | OoD (XXL) | OoD (Large) | OoD (XL) | OoD (XXL) |
|---|---|---|---|---|---|---|---|---|---|
| AdamW | *3.00* | 3.60 | 4.40 | *2.80* | 2.60 | 4.00 | *2.60* | *2.40* | 3.80 |
| $\mu$Adam | 3.40 | *2.20* | *2.20* | 3.00 | *2.40* | *2.40* | 3.20 | 2.60 | *2.60* |
| $VeLO_M$ | 4.60 | 4.00 | 5.00 | 5.40 | 5.40 | 5.80 | 6.00 | 5.40 | 5.80 |
| $LO_M$ | 5.60 | 5.40 | 5.60 | 5.60 | 4.80 | 5.20 | 5.00 | 4.80 | 5.20 |
| $\mu VeLO_M$ (ours) | 2.60 | **1.60** | 1.80 | 2.40 | 2.00 | 2.40 | 2.40 | **1.40** | 2.00 |
| $\mu LO_M$ (ours) | **1.80** | 2.00 | **2.00** | **1.80** | **1.60** | **1.20** | **1.80** | 2.20 | **1.60** |

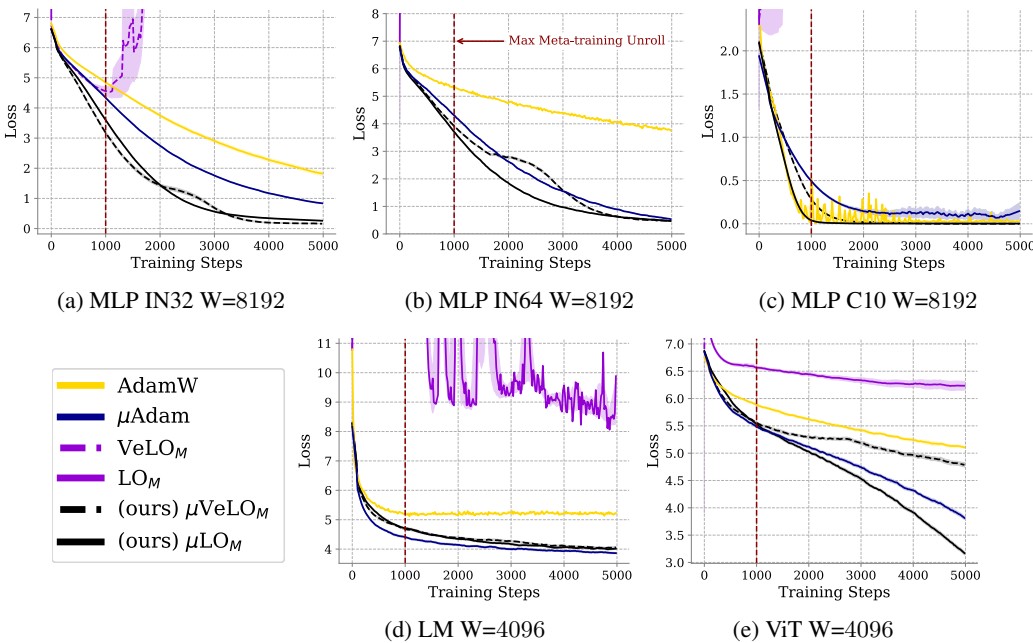

Figure 4: **Evaluating generalization to wider networks for different tasks.** All optimizers are meta-trained or hyperparameter tuned for 1000 inner steps (dotted red line), therefore, any optimization beyond 1000 steps is considered out-of-distribution. We plot average training loss over 5 seeds with standard error bars. We observe that $\mu LO_M$ and $\mu VeLO_M$ generalize smoothly to longer unrolls and all unseen tasks, unlike their SP counterparts which diverge or fail to make progress. $\mu$LOs outperform the extensively tuned AdamW and $\mu$Adam baselines in subfigures (a),(b), match or surpass them in subfigure (c), and exceed or nearly match their performance on far out-of-distribution LM and ViT tasks (subfigures (d) and (e)). Note that all AdamW and $\mu$Adam are tuned on smaller versions of each task, while our $\mu$LOs are only meta-trained on MLP tasks.

**Performance measured by average optimizer rank**   Table 1 reports the average rank of different optimizers on out-of-distribution w.r.t. width tasks (Large (width 2048), XL (width 3072 for transformer and 4096 for MLPs), and XXL (maximum width)). Each entry of the table corresponds to the optimizer's average rank (within the 6 optimizers evaluated) over the 5 tasks in our suite: Cifar 10 MLP image classification, ImageNet 32 MLP image classification, ImageNet 64 MLP image classification, ImageNet 32 ViT image classification, and LM1B transformer language modeling. The optimizers are ranked by their training loss at the given iteration. We report average ranks for 1000 iterations (inner-problem length), 3000 iterations, and 5000 iterations. We **bold** the strongest, underline the second strongest, and *italicize* the third strongest average rank in each column. We observe that, across all iterations and all task sizes (Large, XL, XXL), either $\mu LO_M$ or $\mu VeLO_M$ consistently obtain the best and second-best ranks for all tasks. The per-task-tune hand-designed baselines consistently occupy third and fourth rank, while the SP learned optimizer baselines perform worst, typically failing to optimize at this size. These results demonstrate that meta-training learned optimizers under the $\mu$-parameterization we propose and using our simple meta-training recipe yields substantial improvements in meta-generalization (across various tasks and widths) over SP LOs (previous work) and strong per-task tuned hand-designed baselines.

### 5.2.4 EVALUATING META-GENERALIZATION BEYOND WIDTH

While our main focus is meta-generalization to wider networks While the focus of our paper is improving the meta-generalization of LOs on wider tasks, it is also important to evaluate how these modifications to learned optimizer meta-training impact other axes of generalization. As such, we now study meta-generalization to deeper networks and longer training. While we provide strong AdamW and $\mu$Adam baselines for reference, our focus will be to establish the relative performance $\mu$LOs to SP LOs. Note that $\mu$P theory leveraged by $\mu$LOs specifically concerns transferring hyperparameters to

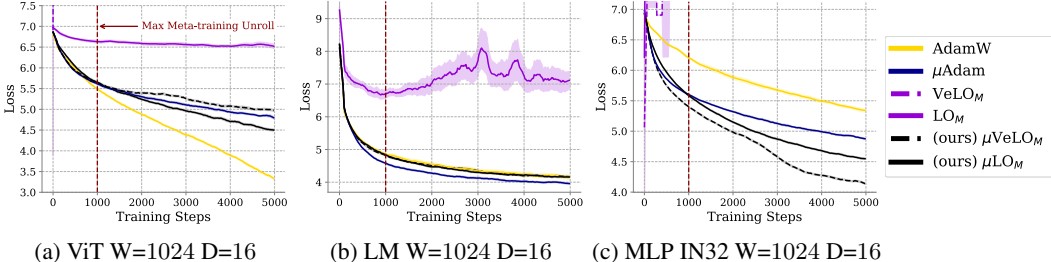

(a) ViT W=1024 D=16     (b) LM W=1024 D=16     (c) MLP IN32 W=1024 D=16

Figure 5: **Evaluating generalization capabilities of $\mu$LOs to deeper networks**. Our focus is on comparing the meta-generalization to deeper tasks of $\mu$LOs to SP LOs (all meta-trained exclusively on MLPs). We also report the performance per-task tuned AdamW and $\mu$Adam for reference. Each plot reports average training loss over 5 seeds with standard error bars. In each case, $\mu$LOs show improved generalization and performance when compared to their SP counterparts.

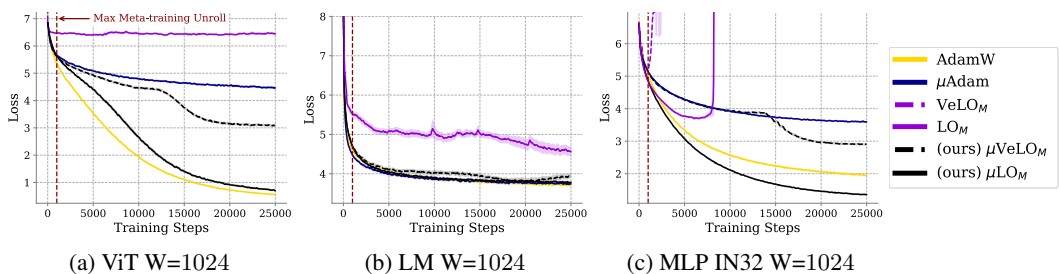

(a) ViT W=1024     (b) LM W=1024     (c) MLP IN32 W=1024

Figure 6: **Evaluating meta-generalization to longer training horizons**. Note that AdamW and $\mu$Adam are evaluated on their tuning tasks here, while LOs are trained on MLPs. We plot average training loss over 5 seeds with standard error bars. We observe that $\mu$LOs seamlessly generalize to training horizons $25\times$ longer than meta-training. In contrast, the best performing SP LO fails to decrease training loss (a), decreases it but suffers instabilities (b), or diverges after 8000 steps (c).

larger-width networks, not longer training horizons or deeper networks. Therefore, any improvements we observe are purely empirical.

**Meta-generalization to deeper networks** In this section, we evaluate LO meta-generalization to deeper networks. Specifically, we increase the number of layers used in MLP, ViT, and LM tasks from 3 to 16, while keeping width=1024 within the range of tuning/meta-training. Figure 5 reports the performance of our learned optimizers on deeper networks. We observe that both $\mu$LO$_M$ and $\mu$VeLO$_M$ optimize stably throughout and generally outperform their counterparts, LO$_M$ and VeLO$_M$, by the end of training on each task, despite being meta-trained on MLPs of exactly the same depth. Moreover, LO$_M$ immediately diverges when optimizing the deep MLP while $\mu$LO$_M$ experiences no instability. Similarly, VeLO$_M$ diverges on ViTs and Transformers, while $\mu$VeLO$_M$ performs well, especially on ViTs. This is remarkable as, unlike width, there is no theoretical justification for $\mu$P's benefit to deeper networks. We hypothesize that $\mu$P's stabilizing effect on the optimizee's activations leads to this improvement in generalization (see Sec. F.1.2 for more details).

**Meta-generalization to longer training** In this subsection, we empirically evaluate the capability of $\mu$LOs to generalize to much longer training horizons than those seen during meta-training. Specifically, we use $\mu$LO$_M$ and LO$_M$ as well as $\mu$VeLO$_M$ and VeLO$_M$ to train three networks with width $w = 1024$: a 3-layer MLP, ViT on $32 \times 32 \times 3$ ImageNet and a 3-layer Transformer for autoregressive language modeling on LM1B. Each model is trained for $25,000$ steps ($25\times$ the longest unroll seen at meta-training time). Figure 6 reports the training loss averaged over 5 random seeds. We observe that $\mu$LO$_M$ and $\mu$VeLO$_M$ stably decrease training loss over time for each task, while LO$_M$ and VeLO$_M$ fail to decrease training loss (a), decreases it but becomes unstable (b), or diverges after 8000 steps (c). While we are uncertain of the exact cause of this improved generalization, we hypothesize that it may be due to the improved pre-activation stability (see Sec. F.1.2 for more details). These results suggest that generalization to longer training horizons is another benefit of using $\mu$LOs.

## 6 LIMITATIONS

We have conducted a systematic empirical study and shown strong results within the scope of our study, there are some limitations of our work. Specifically, (1) we do not meta-train on tasks other than MLPs for image classification, (2) we do not provide an evaluation of models wider than 8192 (MLPs) and 3072/12288 (transformer hidden/FFN size) due to computational constraints in our academic environment, and (3) We did not include an oracle SP AdamW baseline whose hyperparameters are swept at every width due to computational constraints in our academic environment.

## 7 CONCLUSION

We have theoretically and empirically demonstrated that it is possible to obtain a valid $\mu$-parameterization for two state-of-the-art learned optimizer architectures. Under or proposed meta-training recipe, meta-learned optimizers show substantial improvements in meta-generalization properties when compared to strong baselines from previous work. Remarkably, our $\mu$LOs, meta-trained only on MLP tasks, surpass the performance of per-task-tuned hand-designed baselines in terms of average rank on wide OOD tasks. Moreover, our experiments also show that $\mu$LOs meta-trained with our recipe generalize better to wider and, unexpectedly, deeper out-of-distribution tasks than their SP counterparts. When evaluated on much longer training tasks, we observe that $\mu$LOs have a stabilizing effect, enabling meta-generalization to much longer unrolls ($25\times$ maximum meta-training unroll length). All of the aforementioned benefits of $\mu$LOs come at *zero* extra computational cost compared to SP LOs. Our results outline a promising path forward for low-cost meta-training of learned optimizers that can generalize to large unseen tasks.

In future work, it will be important to investigate the benefits of meta-learning optimizers under parameterizations other than $\mu$P that have been shown to admit hyperparameter transfer (Everett et al., 2024). Another important direction of inquiry is to investigate the meta-learning optimizers under parameterizations, like CompleteP (Dey et al., 2025), that have the potential to improve meta-generalization across depth and width. Finally, combining such parameterizations with improved meta-generalization and scalable meta-learning recipes is required for learning truly general-purpose optimizers.

### ACKNOWLEDGMENTS

We acknowledge support from the Mila-Samsung Research Grant, FRQNT New Scholar [*E.B.*], the FRQNT Doctoral (B2X) scholarship [*B.T.*], the Canada CIFAR AI Chair Program [*I.R.*], and the Canada Excellence Research Chairs Program in Autonomous AI [*I.R.*]. We also acknowledge resources provided by Compute Canada, Calcul Québec, and Mila. [*E.O.*] acknowledges funding from PEPR IA (grant SHARP ANR-23-PEIA-0008). He was granted access to the AI resources of IDRIS under the allocation 2025- AD011015884R1.

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
