## A  PROOF OF PROPOSITION 4.1

For the reader's convenience, we will first review the input, output, update, and scaling of the per-parameter `small_fc_lopt` Metz et al. (2022a) learned optimizer as it is necessary background for understanding our proof. This corresponds to the architecture of the $\mu\text{LO}_M$, $\mu\text{LO}_S$, $\text{LO}_M$, and $\text{LO}_S$ optimizers used throughout our experiments. In section A.1.2, we will also review the input, output, update, and scaling of VeLO, the architecture used for $\mu\text{VeLO}_M$ and $\text{VeLO}_M$. Note that the VeLO Metz et al. (2022b) architecture uses an almost-identical `small_fc_lopt` network to produce per-parameter updates. The main difference is that VeLO uses an LSTM to generate the parameters of `small_fc_lopt` for each tensor in the network at each optimization step.

### A.1  $\mu\text{LO}_M$ AND $\mu\text{VeLO}_M$ INPUT, OUTPUT, UPDATE, AND SCALING.

#### A.1.1  THE SMALL_FC_LOPT ARCHITECTURE

`small_fc_lopt` maintains three different per-parameter momentum accumulators ($M_{t,i}$) and one variance accumulator ($V_t$). In addition, it also maintains six adafactor-style accumulators of the column-wise ($c_{t,i}$) and row-wise ($r_{t,i}$) mean of the squared gradient. The accumulator update is given as follows:

$$M_{t,i} = \beta_i M_{t-1,i} + (1 - \beta_i)\nabla_t \qquad\qquad i \in \{1, 2, 3\},$$
$$V_t = \beta_4 V_{t-1} + (1 - \beta_4)\nabla_t^2,$$
$$r_{t,i} = \beta_i r_{t-1,i} + (1 - \beta_i)\,\texttt{row\_mean}(\nabla_t^2), \qquad\qquad i \in \{5, 6, 7\},$$
$$c_{t,i} = \beta_i c_{t-1,i} + (1 - \beta_i)\,\texttt{col\_mean}(\nabla_t^2), \qquad\qquad i \in \{5, 6, 7\},$$
$$U_t := [M_{t,1}, M_{t,2}, M_{t,3}, V_t, r_{t,5}, r_{t,6}, r_{t,7}, c_{t,5}, c_{t,6}, c_{t,7}].$$

Here, we slightly abuse notation and define $U_t$ to be the entire accumulator state for all parameters in the optimizee (column-wise and row-wise features are repeated for notational convenience). After updating these accumulators, `small_fc_lopt` computes additional learned optimizer input features:

$$F_i^{(\nabla)} = \nabla_t \odot \sqrt{\frac{\frac{1}{m}\sum_{h=1}^m (r_{t,i})_h}{r_{t,i}c_{t,i}^T}},$$

$$F_i^{(M)} = M_{t,j} \odot \sqrt{\frac{\frac{1}{m}\sum_{h=1}^m (r_{t,i})_h}{r_{t,i}c_{t,i}^T}},$$

$$R_t = \left[\frac{1}{\sqrt{r_{t,5}}}, \frac{1}{\sqrt{r_{t,6}}}, \frac{1}{\sqrt{r_{t,7}}}, \frac{1}{\sqrt{c_{t,5}}}, \frac{1}{\sqrt{c_{t,6}}}, \frac{1}{\sqrt{c_{t,7}}}, \frac{M_{t,1}}{\sqrt{v}}, \frac{M_{t,2}}{\sqrt{v}}, \frac{M_{t,3}}{\sqrt{v}}, \frac{1}{\sqrt{v}}\right],$$

$$H_t = \left[F_1^{(\nabla)}, F_2^{(\nabla)}, F_3^{(\nabla)}, F_1^{(M)}, F_2^{(M)}, F_3^{(M)}\right],$$

$$\hat{A}_t = \theta_t \odot \nabla_t \odot H_t \odot R_t \odot U_t.$$

Where $\odot$ denotes matrix concatenation across the feature dimension, $\theta_t$ are the optimizee's parameters, $\nabla_t$ is the optimizee's gradient, $H_t$ are adafactor normalized features, and $R_t$ are reciprocal features. Note that $\hat{A}_t \in \mathbb{R}^{|\theta|\times 28}$. The features within a parameter tensor are now normalized by their RMS-norm. Let $W^{(j)} \in \mathbb{R}^{m\times n}$ be the optimizee's j'th tensor and take $\hat{A}^{(j)} \in \mathbb{R}^{mn\times 28}$ to be the features of this tensor at timestep $t$. Each feature $i$ within $\hat{A}^{(j)}$ is then then normalized as follows:

$$\bar{A}_{:,i}^{(j)} = \frac{\hat{A}_{:,i}^{(j)}}{\sqrt{\frac{1}{mn}\sum_{h=1}^{mn}(\hat{A}_{h,i}^{(j)})^2}}. \tag{4}$$

Finally, the normalized features $\bar{A}$ are concatenated with timestep embeddings from step $t$ to form the complete input features for `small_fc_lopt`:

$$T_t = [\tanh\left(\frac{t}{x}\right)\text{ for }x \in \{1, 3, 10, 30, 100, 300, 1000, 3000, 10000, 30000, 100000\}],$$
$$A_t = \bar{A}_t \odot T_t.$$

Concretely, in Metz et al. (2022a), `small_fc_lopt`'s architecture is a two-hidden-layer 4 hidden-dimension MLP with ReLU activations: $f_\phi(\boldsymbol{A}) = \boldsymbol{W}_2(ReLU(\boldsymbol{W}_1 ReLU(\boldsymbol{W}_0 \boldsymbol{A} + \boldsymbol{b}_0) + \boldsymbol{b}_1) + \boldsymbol{b}_2$. At each step, the learned optimizer maps the input features for each parameter, $p$, in the optimizee to a two-dimensional vector, $[d, m]$. At step $t$, the learned optimizer update for all parameters $p$ is given as follows:

$$f_\phi(\boldsymbol{A}_p) = [d_p, m_p];$$
$$p_t = p_{t-1} - \lambda_1 d_p e^{(\lambda_2 m_p)}. \tag{5}$$

Where $\lambda_1 = \lambda_2 = 0.001$ to bias initial steps towards being small. We will now show that the inputs to `small_fc_lopt` scales like $\Theta(1)$ as $n \to \infty$. Let's first see that any RMS-normalized quantity (e.g., the input to `small_fc_lopt`) is $\Theta(1)$, which we will subsequently use in our proof of propositions 4.1 and 4.2.

**Definition A.1.** Let $\boldsymbol{W} \in \mathbb{R}^{m \times n}$ be the weight matrix of a neural network. Let $\boldsymbol{v} \in \mathbb{R}^{mn}$ be a vector, whose entries are statistics of parameters in $\boldsymbol{W}$. We call

$$\bar{\boldsymbol{v}} = \frac{\boldsymbol{v}}{\text{RMS}(\boldsymbol{v})} \quad ; \quad \text{RMS}(\boldsymbol{v}) = \sqrt{\frac{1}{mn}} \|\boldsymbol{v}\|_2. \tag{6}$$

The RMS-normalized Zhang & Sennrich (2019) version of $\boldsymbol{v}$.

**Proposition A.2.** *Let $\boldsymbol{v} \in \mathbb{R}^{mn}$ be a vector whose entries scale like $\Theta(f(n))$, where $f : \mathbb{R} \to \mathbb{R}$ is a continuous function. Then, the entries of the RMS-normalized counterpart of $\boldsymbol{v}$, $\bar{\boldsymbol{v}} \in \mathbb{R}^{mn}$ will scale like $\Theta(1)$.*

*Proof.* Let $\boldsymbol{v} \in \mathbb{R}^{mn}$ be a vector and $\bar{\boldsymbol{v}} \in \mathbb{R}^{mn}$ denote its RMS-normalized counterpart. Then,

$$\bar{\boldsymbol{v}} = \frac{\boldsymbol{v}}{\sqrt{\frac{1}{mn} \sum_{h=1}^{mn} \boldsymbol{v}_h^2}} \tag{7}$$

where the division is elementwise. From the definition of $\Theta$, we know there exist constants $c_1, c_2 > 0$ and $N \in \mathbb{N}$ such that for all $n \geq N$ and every $h \in \{1, \dots, mn\}$,

$$c_1 |f(n)| \leq |v_h| \leq c_2 |f(n)|.$$

Thus we have:

$$v_h^2 \in \left[ c_1^2 f(n)^2, \ c_2^2 f(n)^2 \right],$$
$$\sum_{h=1}^{mn} v_h^2 \in \left[ mn \, c_1^2 f(n)^2, \ mn \, c_2^2 f(n)^2 \right].$$
$$\frac{1}{mn} \sum_{h=1}^{mn} v_h^2 \in \left[ c_1^2 f(n)^2, \ c_2^2 f(n)^2 \right],$$
$$\sqrt{\frac{1}{mn} \sum_{h=1}^{mn} v_h^2} \in \left[ c_1 |f(n)|, \ c_2 |f(n)| \right] = \Theta\big(f(n)\big). \tag{8}$$

Since both numerator and denominator of 7 are $\Theta(f(n))$, their ratio is $\bar{v}_h = \Theta(1)$ for each $h$. This completes the proof. $\square$

**Corollary A.3.** *Assuming that time features are independent of width $n$, the coordinates of the input features to `small_fc_lopt`, as defined above, are $\Theta(1)$ as $n \to \infty$.*

*Proof.* This follows directly from proposition A.2 since all non-time features in `small_fc_lopt` are RMS-normalized. $\square$

A.1.2   THE VELO ARCHITECTURE

**VeLO** uses an LSTM hypernetwork to produce the parameters, $\phi_W$, of a `small_fc_lopt` optimizer for each weight matrix $W$ in the optmizee network. Therefore, VeLO has the same accumulators as `small_fc_lopt`. VeLO's LSTM also outputs a learning rate multiplier, $\alpha_W$. For a parameter $p$ of $W$, the update becomes:

$$f_{\phi_W}(A_p^*) = [d_p, m_p];$$
$$p_t = p_{t-1} - \alpha_W \lambda_1 d_p e^{(\lambda_2 m_p)}. \tag{9}$$

Where $A_p^*$ is a slightly modified version of the features outlined in the previous section (see Tab. 3 for details), crucially, the features $A_p^*$ are all RMS-normalized as illustrated in the previous section.

To produce $\phi_W$ and $\alpha_W$, VeLO's LSTM takes as input 9 remaining time features ($T$), 9 EMA loss features ($L$), a one-hot vector representing the tensor's rank, three momentum features (`var_mom`$_k$ for $k \in \{1, 2, 3\}$), and two variance features (`mean_rms var_rms`). For our goal of understanding valid parameterizations for VeLO, the most important LSTM features are the variance and momentum features as they are the only features that require further analysis of width scaling:

$$\hat{m}_k = \frac{1}{mn} \sum_i^m \sum_j^n \frac{M_{i,j}^{(k)}}{\text{RMS}(W)},$$

$$\texttt{var\_mom}_k = c_1 \; \text{clip}\Big(\log\Big[\frac{c_2}{mn} \sum_{i,j} \Big(\frac{M_{ij}^{(k)}}{\text{RMS}(W)} - \hat{m}_k\Big)^2\Big], -\tau, \tau\Big),$$

$$\texttt{mean\_rms} = c_1 \; \text{clip}\Big(\log\Big[\frac{c_2}{mn} \sum_{i,j} \frac{V_{ij}}{\text{RMS}(W)}\Big], -\tau, \tau\Big), \; \text{and}$$

$$\texttt{var\_rms}_k = c_1 \; \text{clip}\Big(\log\Big[\frac{c_2}{mn} \sum_{i,j} \Big(\frac{V_{ij}}{\text{RMS}(W)} - \hat{m}_k\Big)^2\Big], -\tau, \tau\Big).$$

Where we set $c_1 = \frac{1}{2}, c_2 = 10$, and $\tau = 5$ following Metz et al. (2022b). Note that, in general, the quantities calculated within the log may not be nicely bounded, but since these features are clipped, straightforward analysis shows these features are $\Theta(1)$.

**Proposition A.4.** *Let $W \in \mathbb{R}^{m \times n}$ be a weight matrix whose entries scale as $\Theta(n^p)$. Let $\hat{m}_k$, `var_mom`$_k$, `mean_rms`, and `var_rms`$_k$ be defined as above. Assume $M^{(k)}$ has the same per-entry scaling as $W$, and $V$ has entries scaling as $\Theta(n^{2p})$. Then each of $\hat{m}_k$, `var_mom`$_k$, `mean_rms`, and `var_rms`$_k$ is $\Theta(1)$ as $n \to \infty$.*

*Proof.* First, observe that

$$\text{RMS}(W) = \sqrt{\frac{1}{mn} \sum_{i,j} W_{i,j}^2} = \sqrt{\Theta(n^{2p})} = \Theta(n^p).$$

Since $M_{i,j}^{(k)} = \Theta(n^p)$, it follows that

$$\frac{M_{i,j}^{(k)}}{\text{RMS}(W)} = \Theta(n^p/n^p) = \Theta(1).$$

Hence

$$\hat{m}_k = \frac{1}{mn} \sum_{i,j} \Theta(1) = \Theta(1).$$

Next, consider the argument of the logarithm in `var_mom`$_k$:

$$\frac{c_2}{mn} \sum_{i,j} \Big(\frac{M_{i,j}^{(k)}}{\text{RMS}(W)} - \hat{m}_k\Big)^2.$$

Each term $\frac{M_{i,j}^{(k)}}{\text{RMS}(\boldsymbol{W})} - \hat{m}_k$ is the difference of two $\Theta(1)$ quantities, hence $\Theta(1)$. Summing $mn$ such terms and dividing by $mn$ yields $\Theta(1)$. Thus

$$\log\left[\frac{c_2}{mn}\sum_{i,j}\left(\frac{M_{i,j}^{(k)}}{\text{RMS}(\boldsymbol{W})} - \hat{m}_k\right)\right] = \Theta(1),$$

and clipping to $[-\tau, \tau]$ gives $\Theta(1)$. Multiplying by the constant $c_1$ preserves $\Theta(1)$. Therefore `var_mom`$_k = \Theta(1)$.

For `mean_rms`, note $V_{i,j} = \Theta(n^{2p})$, so

$$\frac{V_{i,j}}{\text{RMS}(\boldsymbol{W})} = \Theta\big(n^{2p}/n^p\big) = \Theta(n^p).$$

Hence

$$\frac{c_2}{mn}\sum_{i,j}\frac{V_{i,j}}{\text{RMS}(\boldsymbol{W})} = \Theta(n^p),$$

and

$$\log\big[\Theta(n^p)\big] = \Theta(\log n).$$

Clipping $\log(n^p)$ to $[-\tau, \tau]$ yields a bounded constant $\Theta(1)$, and multiplication by $c_1$ gives `mean_rms` $= \Theta(1)$.

Finally, for `var_rms`$_k$, we have

$$\frac{V_{i,j}}{\text{RMS}(\boldsymbol{W})} - \hat{m}_k = \Theta(n^p) - \Theta(1) = \Theta(n^p),$$

so

$$\left(\frac{V_{i,j}}{\text{RMS}(\boldsymbol{W})} - \hat{m}_k\right)^2 = \Theta(n^{2p}).$$

Summing over $mn$ entries and dividing by $mn$ yields $\Theta(n^{2p})$. Taking the logarithm gives $\Theta(\log n)$, clipping to $[-\tau, \tau]$ yields $\Theta(1)$, and multiplying by $c_1$ preserves $\Theta(1)$. Hence `var_rms`$_k = \Theta(1)$, completing the proof. $\qquad\square$

**Corollary A.5.** *Assuming that time features are independent of width n, the coordinates of the input features to* `VeLO`*'s LSTM, as defined above, are* $\Theta(1)$ *as* $n \to \infty$.

*Proof.* This follows directly from proposition A.4 since all the other input features in `VeLO` trivially $\Theta(1)$ as $n \to \infty$. $\qquad\square$

### A.2 PROOF: $\mu$-PARAMETERIZATION FOR LEARNED OPTIMIZERS

For the reader's convenience, we will now restate the $\mu$P desiderata (Appendix J.2 Yang et al. (2022)) which will be used by our proof. When using a maximal update parameterization, at any point during training, the following conditions should be met:

1. **(Activation Scale)** Every (pre-)activation vector $x \in \mathbb{R}^n$ in the network should have $\Theta(1)$-sized coordinates.

2. **(Output Scale)** The output of the neural network $f_\theta(x)$ should be $O(1)$.

3. **(Maximal Updates)** All parameters should be updated as much as possible without divergence. In particular, updates should scale in width so that each parameter has nontrivial dynamics in the infinite-width limit.

While we do not go into the level of mathematical detail of Yang & Littwin (2023), our intention in propositions 4.1 and 4.2 is to show that the above desiderata are satisfied in practice by the two popular learned optimizer architectures we study.

**Proposition 4.1.** *Assume that the Learned Optimizer $f_\phi$ has the form* $small\_fc\_lopt$ *is fed with features given in Appendix A.1.1 and that during training the optimizee's parameters and input data*

*become aligned leading to Law of Large Numbers (LLN) scaling, then the update, initialization, and pre-activation multiplier above is sufficient to obtain a Maximal Update Parametrization.*

**Proposition 4.2.** *Assume that $\phi$ in Proposition 4.1 is generated using an LSTM with the input features described in Appendix A.1.2 and that during training the optimizee's parameters and input data become aligned leading to Law of Large Numbers (LLN) scaling, then the update, initialization, and pre-activation multiplier above is sufficient to obtain a Maximal Update Parametrization.*

*Proof.* We will now prove both statements by arguing that, in each case, the update of $f_\phi$ is in $\Theta(1)$, implying that our parameterization is correct. Without loss of generality, we will assume that the optimizee network has input dimension $d$, hidden dimension $n$ (width), and output dimension $c$. Let $\boldsymbol{W}$ be some weight matrix in the optimizee network, let the update produced by $f_\phi$ be $\Delta\boldsymbol{W}$ and let $\boldsymbol{A}$ be the corresponding input features such that $\Delta\boldsymbol{W} = f_\phi(\boldsymbol{A})$.

- In the case of `small_fc_lopt`, $f_\phi(\boldsymbol{x}) = \Theta(1)$ since its input features, $\boldsymbol{A}$, are $\Theta(1)$ due to normalization (see corollary A.3).

- In the case of VeLO, we must also show that the LSTM hypernetwork does not introduce additional dependence on the width, $n$. From corollary A.5 we know that the LSTM hypernetwork will produce parameters, $\phi_{\boldsymbol{W}}$, of `small_fc_lopt` and an LR multiplier, $\alpha_{\boldsymbol{W}}$ which are $\Theta(1)$ since all inputs to the LSTM are $\Theta(1)$. Therefore, $f_\phi(\boldsymbol{x}) = \Theta(1)$ for VeLO aswell.

This fact is henceforth referred to as property (A). We will assume that the optimizee network follows our proposed $\mu$-parameterization from Sec. 4, and show that we satisfy the desiderata of $\mu$P (outlined above) for any weight layer, $\boldsymbol{W}$, in the network. Concretely, we will show that for input and hidden layers,

$$\boldsymbol{x}_i = \Theta(1) \Rightarrow (\boldsymbol{W}\boldsymbol{x})_i = \Theta(1) \text{ and } ((\boldsymbol{W} + \Delta\boldsymbol{W})\boldsymbol{x})_i = \Theta(1) \tag{10}$$

that for the output layer

$$\boldsymbol{x}_i = \Theta(1) \Rightarrow (\boldsymbol{W}\boldsymbol{x})_i = O(1) \text{ and } ((\boldsymbol{W} + \Delta\boldsymbol{W})\boldsymbol{x})_i = O(1) \tag{11}$$

and that for all layers

$$(\Delta\boldsymbol{W}\boldsymbol{x})_i = \Theta(1). \tag{12}$$

Statements 10, 11, and 12 correspond to desiderata (1) activation scale, (2) output scale, and (3) maximal updates, respectively. In satisfying these statements, we will show that our parameterization is indeed a maximal update parameterization.

**Output weights.** Here, the input $\boldsymbol{x}$ has $\Theta(1)$ coordinates, we initialize the output matrix $\boldsymbol{W}$ with entries of variance 1 (which is necessary) and rescale the logits with $1/n$. Therefore, the output, $(1/n)\boldsymbol{W}\boldsymbol{x}$, is $O(1)$ by the LLN (**Output Scale Property**). From property (A), we know that $\Delta\boldsymbol{W} = f_\phi(\nabla\boldsymbol{W})$ has coordinates in $\Theta(1)$, so the entries of $\boldsymbol{W} + \Delta\boldsymbol{W}$ still have variance 1 and $\frac{1}{n}((\boldsymbol{W} + \Delta\boldsymbol{W})\boldsymbol{x})_i$ is $O(1)$. Moreover, $\frac{1}{n}(\Delta\boldsymbol{W}\boldsymbol{x})_i = \Theta(1)$ by LLN (**Maximal updates**).

**Hidden weights.** Since hidden weights are initialized with variance $1/n$ and $\boldsymbol{x}_i = \Theta(1)$, the coordinates of $\boldsymbol{W}\boldsymbol{x}$ are $\Theta(1)$ by LLN. From property (A), we know that $f_\phi(\boldsymbol{A}) = \Theta(1)$. Therefore, to ensure $\Delta\boldsymbol{W} \cdot \boldsymbol{x}$ is coordinate-wise bounded, we must re-scale the parameter updates:

$$\Delta\boldsymbol{W} = \frac{1}{n}f_\phi(\boldsymbol{A}).$$

Since this rescaling implies that $\Delta\boldsymbol{W}$ is $\Theta(1/n)$, the entries of $\boldsymbol{W} + \Delta\boldsymbol{W}$ still scale like $1/n$ and $((\boldsymbol{W} + \Delta\boldsymbol{W})\boldsymbol{x})_i$ is $\Theta(1)$. Moreover, since $\Delta\boldsymbol{W}$ is $\Theta(1/n)$ and $\boldsymbol{x}_i = \Theta(1)$, then $1/n(\Delta\boldsymbol{W}\boldsymbol{x})_i = \Theta(1)$ by LLN (**Maximal updates**).

**Input weights.** Recall that $d$, the input dimension, is fixed and does not grow with $n$. Since the input $\boldsymbol{x}_i = \Theta(1)$ and $\boldsymbol{W}$ has entries with variance $1/d$ in $\Theta(1)$, then the coordinates of pre-activation $\boldsymbol{W}\boldsymbol{x}$ are $\Theta(1)$. From property (A), we know that $f_\phi(\boldsymbol{A}) = \Theta(1)$. Therefore, $\Delta\boldsymbol{W}$ is $\Theta(1)$, the entries of $\boldsymbol{W} + \Delta\boldsymbol{W}$ still have $\Theta(1)$ coordinates and $((\boldsymbol{W} + \Delta\boldsymbol{W})\boldsymbol{x})_i$ is $\Theta(1)$ (as $d$ is fixed). Moreover, $\Delta\boldsymbol{W}\boldsymbol{x}$ will have coordinate sizes that depend on the input dimension, $d$, but not the width. Therefore, $(\Delta\boldsymbol{W}\boldsymbol{x})_i = \Theta(1)$(**Maximal updates**). $\square$

### A.3 SUMMARY OF LEARNED OPTIMIZER INPUT FEATURES

The following section contains easy-to-read tables which report the exact learned optimizer input features for `small_fc_lopt` (Table 2) and `VeLO` (Tables 3 and 4). The tables also report the entry-wise scaling of the features before RMS-normalization and the number of features of each type. Entry-wise scaling is reported assuming a hidden weight matrix. The original implementation of these optimizers along with features calculation can be accessed here[2].

---

[2]`https://github.com/google/learned_optimization/blob/main/learned_ optimization/learned_optimizers/adafac_mlp_lopt.py` and `https://github. com/google/learned_optimization/blob/main/learned_optimization/research/ general_lopt/hyper_v2.py`

Table 2: $\mu\textbf{P}$ **scaling for hidden layers of per-parameter features input to** $\mu\textbf{LO}_M$. All the coefficients, $\beta_i$, are learnable parameters adjusted during meta-optimization. All feature calculations and scalings are reported for a hidden weight matrix $\boldsymbol{W} \in \mathbb{R}^{m \times n}$ in an optimizee network following our proposed $\mu$-parameterization. Here, $n$ is the width and $m = kn$ for some constant $k \in \mathbb{R}$. In this case, the entries of the gradient of $\boldsymbol{W}$, $\nabla_t$, scale like $\Theta(\frac{1}{n})$, where $n$ is the width of the model. **Notation.** The table will use $\nabla_{t,i}$ or $\nabla_{t,j}$ to indicate the variable's dependence on time $t$ and coefficient $\beta_i$ or $\beta_j$, respectively. $(\nabla_{t,j})_{r,c}$ will designate indexing into row $r$ and column $c$ of the quantity $\nabla_{t,j}$. **DISCLAIMER: All features in our tables report scaling before the RMS-normalization.**

| Type | # | Description | Accumulator Update/Equation | Scaling |
|---|---|---|---|---|
| **Accumulators** | 3 | Momentum accumulators with coefficients $\beta_i, i \in \{1,2,3\}$. | $\boldsymbol{M}_{t,i} = \beta_i \boldsymbol{M}_{t-1,i} + (1-\beta_i)\nabla_t$ | $\Theta(\frac{1}{n})$ |
| | 1 | Second moment accumulator with coefficient $\beta_4$. | $\boldsymbol{V}_t = \beta_4 \boldsymbol{V}_{t-1} + (1-\beta_4)\nabla_t^2$ | $\Theta(\frac{1}{n^2})$ |
| | 3 | Adafactor row accumulator with coefficients $\beta_i, i \in \{5,6,7\}$. | $\boldsymbol{r}_{t,i} = \beta_i \boldsymbol{r}_{t-1,i} + (1-\beta_i)\,\texttt{row\_mean}(\nabla_t^2)$ | $\Theta(\frac{1}{n^2})$ |
| | 3 | Adafactor accumulator with coefficients $\beta_i, i \in \{5,6,7\}$. | $\boldsymbol{c}_{t,i} = \beta_i \boldsymbol{c}_{t-1,i} + (1-\beta_i)\,\texttt{col\_mean}(\nabla_t^2)$ | $\Theta(\frac{1}{n^2})$ |
| **Accumulator Features** | 3 | Momentum values normalized by the square root of the second moment for $i \in \{5,6,7\}$. | $\dfrac{\boldsymbol{M}_{t,i}}{\sqrt{\boldsymbol{V}_t}}$ | $\Theta(1)$ |
| | 1 | The reciprocal square root of the second moment value. | $\dfrac{1}{\sqrt{\boldsymbol{V}}}$ | $\Theta(n)$ |
| | 6 | The reciprocal square root of the Adafactor accumulators. | $\dfrac{1}{\sqrt{\boldsymbol{r}_{t,i}}}$ OR $\dfrac{1}{\sqrt{\boldsymbol{c}_{t,i}}}$ | $\Theta(n)$ |
| | 3 | Adafactor gradient features for $i \in \{5,6,7\}$. | $\nabla_t \cdot \sqrt{\dfrac{\frac{1}{m}\sum_{h=1}^{m}(\boldsymbol{r}_{t,i})_h}{\boldsymbol{r}_{t,i}\boldsymbol{c}_{t,i}^T}}$ | $\Theta(1)$ |
| | 3 | Adafactor momentum features for $i,j \in \{(5,1),(6,2),(7,3)\}$. | $\boldsymbol{M}_{t,j} \cdot \sqrt{\dfrac{\frac{1}{m}\sum_{h=1}^{m}(\boldsymbol{r}_{t,i})_h}{\boldsymbol{r}_{t,i}\boldsymbol{c}_{t,i}^T}}$ | $\Theta(1)$ |
| **Time Features** | 11 | Time Features for $x \in \{1,3,10,30,100,300,1000,3000,10^4,3\cdot10^4,10^5\}$. | $\tanh\left(\frac{t}{x}\right)$ | $\Theta(1)$ |
| **Parameters** | 1 | Parameter value. | $\boldsymbol{W}_t$ | $\Theta(\frac{1}{n})$ |
| | 1 | Gradient value. | $\nabla_t$ | $\Theta(\frac{1}{n})$ |
| **Total** | 39 | – | – | – |

Table 3: $\mu$**P scaling of per-parameter features input to the per-parameter network of $\mu$VeLO$_M$.** All feature calculations and scalings are reported for a hidden weight matrix $\boldsymbol{W} \in \mathbb{R}^{m \times n}$ in an optimizee network following our proposed $\mu$-parameterization. Here, $n$ is the width and $m = kn$ for some constant $k \in \mathbb{R}$. In this case, the entries of the gradient of $\boldsymbol{W}$, $\nabla_t$, scale like $\Theta(\frac{1}{n})$, where $n$ is the width of the model. **Notation.** The table will use $\nabla_{t,i}$ or $\nabla_{t,j}$ to indicate the variable's dependence on time $t$ and coefficient $\beta_i$ or $\beta_j$, respectively. $(\nabla_{t,j})_{r,c}$ will designate indexing into row $r$ and column $c$ of the quantity $\nabla_{t,j}$. **DISCLAIMER: All features in our tables report scaling before the RMS-normalization.**

| Type | # | Description | Accumulator Update/Equation | Scaling |
|---|---|---|---|---|
| **Accumulators** | 3 | Momentum accumulators with coefficients $\beta_i, i \in \{1,2,3\}$. | $\boldsymbol{M}_{t,i} = \beta_i \boldsymbol{M}_{t-1,i} + (1-\beta_i)\nabla_t$ | $\Theta(\frac{1}{n})$ |
| | 1 | Second moment accumulator with coefficient $\beta_4$. | $\boldsymbol{V}_t = \beta_4 \boldsymbol{V}_{t-1} + (1-\beta_4)\nabla_t^2$ | $\Theta(\frac{1}{n^2})$ |
| | 3 | Adafactor row accumulator with coefficients $\beta_i, i \in \{5,6,7\}$. | $\boldsymbol{r}_{t,i} = \beta_i \boldsymbol{r}_{t-1,i} + (1-\beta_i)\,\texttt{row\_mean}(\nabla_t^2)$ | $\Theta(\frac{1}{n^2})$ |
| | 3 | Adafactor accumulator with coefficients $\beta_i, i \in \{5,6,7\}$. | $\boldsymbol{c}_{t,i} = \beta_i \boldsymbol{c}_{t-1,i} + (1-\beta_i)\,\texttt{col\_mean}(\nabla_t^2)$ | $\Theta(\frac{1}{n^2})$ |
| **Accumulator Features** | 3 | Momentum values normalized by the square root of the second moment for $i \in \{5,6,7\}$. | $\dfrac{\boldsymbol{M}_{t,i}}{\sqrt{\boldsymbol{V}_t}}$ | $\Theta(1)$ |
| | 1 | The reciprocal square root of the second moment value. | $\dfrac{1}{\sqrt{\boldsymbol{V}}}$ | $\Theta(n)$ |
| | 6 | The reciprocal square root of the Adafactor accumulators. | $\dfrac{1}{\sqrt{\boldsymbol{r}_{t,i}}}$ OR $\dfrac{1}{\sqrt{\boldsymbol{c}_{t,i}}}$ | $\Theta(n)$ |
| | 3 | Adafactor gradient features for $i \in \{5,6,7\}$. | $\nabla_t \cdot \sqrt{\dfrac{\frac{1}{m}\sum_{h=1}^m (\boldsymbol{r}_{t,i})_h}{\boldsymbol{r}_{t,i}\boldsymbol{c}_{t,i}^T}}$ | $\Theta(1)$ |
| | 3 | Adafactor momentum features for $i,j \in \{(5,1),(6,2),(7,3)\}$. | $\boldsymbol{M}_{t,j} \cdot \sqrt{\dfrac{\frac{1}{m}\sum_{h=1}^m (\boldsymbol{r}_{t,i})_h}{\boldsymbol{r}_{t,i}\boldsymbol{c}_{t,i}^T}}$ | $\Theta(1)$ |
| **Parameters** | 1 | Parameter value. | $\boldsymbol{W}_t$ | $\Theta(\frac{1}{n})$ |
| | 1 | Gradient value. | $\nabla_t$ | $\Theta(\frac{1}{n})$ |
| | 1 | Gradient value. | $\texttt{clip}(\nabla_t, -0.1, 0.1)$ | $\Theta(\frac{1}{n})$ |
| **Total** | 29 | – | – | – |

Table 4: **Per-tensor features used as input to VeLO's LSTM.** All feature calculations and scalings are reported for a hidden weight matrix $W \in \mathbb{R}^{m \times n}$ in an optimizee network following our proposed $\mu$-parameterization. Here, $n$ is the width and $m = kn$ for some constant $k \in \mathbb{R}$. In this case, the entries of the gradient of $W$, $\nabla_t$, scale like $\Theta(\frac{1}{n})$, where $n$ is the width of the model.

| Type | # | Description | Equation | Scaling |
|---|---|---|---|---|
| **Accumulator Features** | 3 | Variance across coordinates of the 3 momentum accumulator matrices normalized by the RMS of the current parameter values $i \in \{1, 2, 3\}$ | `var_mom`$_i$ (Sec. A.1.2) | $\Theta(1)$ |
| | 1 | Mean across coordinates of variance accumulator normalized by the parameter RMS | `mean_rms` (Sec. A.1.2) | $\Theta(1)$ |
| | 3 | Coordinate-wise mean of the variance accumulator. $i \in \{1, 2, 3\}$ | `var_rms`$_i$ (Sec. A.1.2) | $\Theta(1)$ |
| **Tensor Rank** | 5 | A one hot vector representing the tensor's rank, $r$. | $e_r$ | $\Theta(1)$ |
| **EMA Loss Features** | 9 | EMAs of the loss at different timescales chosen based on the number of steps. Values are normalized by the max and min losses seen so far. | see Metz et al. (2022b) | $\Theta(1)$ |
| **Remaining Time Features** | 9 | Time Features for $x \in \{0.03, 0.1, 0.2, 0.4, 0.6, 0.8, 0.9, 1.0, 1.1\}$. | $\tanh(t/T - 10x)$ | $\Theta(1)$ |
| **Total** | 30 | – | – | – |

Table 5: **Meta-training and hyperparameter configurations of LOs and baselines in our empirical evaluation.** The small_fc_lopt and VeLO architectures were initially proposed in (Metz et al., 2022a) and (Metz et al., 2022b). See Tab. 10 for a list of all tasks used in this work.

| Identifier | Type | Architecture | Optimizee Par. | Meta-Training / Tuning Task(s) |
|---|---|---|---|---|
| $\mu$LO$_S$ | Ours | small_fc_lopt | $\mu$LO Sec. 4 | IN32$\mathcal{T}^{\text{MLP}}_{(3,128)}$ |
| $\mu$LO$_M$ | Ours | small_fc_lopt | $\mu$LO Sec. 4 | IN32$\mathcal{T}^{\text{MLP}}_{(3,128)}$,IN32$\mathcal{T}^{\text{MLP}}_{(3,512)}$,IN32$\mathcal{T}^{\text{MLP}}_{(3,1024)}$ |
| $\mu$VeLO$_M$ | Ours | VeLO | $\mu$LO Sec. 4 | IN32$\mathcal{T}^{\text{MLP}}_{(3,128)}$,IN32$\mathcal{T}^{\text{MLP}}_{(3,512)}$,IN32$\mathcal{T}^{\text{MLP}}_{(3,1024)}$ |
| LO$_S$ | LO Baseline | small_fc_lopt | SP | IN32$\mathcal{T}^{\text{MLP}}_{(3,128)}$ |
| LO$_M$ | LO Baseline | small_fc_lopt | SP | IN32$\mathcal{T}^{\text{MLP}}_{(3,128)}$,IN32$\mathcal{T}^{\text{MLP}}_{(3,512)}$,IN32$\mathcal{T}^{\text{MLP}}_{(3,1024)}$ |
| VeLO$_M$ | LO Baseline | VeLO | SP | IN32$\mathcal{T}^{\text{MLP}}_{(3,128)}$,IN32$\mathcal{T}^{\text{MLP}}_{(3,512)}$,IN32$\mathcal{T}^{\text{MLP}}_{(3,1024)}$ |
| VeLO-4000 | Oracle LO Baseline | VeLO | SP | See Metz et al. (2022b) (Appendix C.2) |
| $\mu$Adam | Baseline | – | $\mu$P Adam | per-task tuning (see Tab. 7) |
| AdamW | Baseline | – | SP | per-task tuning (see Tab. 9) |

## B  HAND DESIGNED OPTIMIZER HYPERPARAMETER TUNING

To provide strong baselines for our study, we tune the hyperparameters of AdamW and $\mu$Adam for more than 500 trials on one instance of each task in our evaluation suite. Since the largest width task seen by $\mu$LO$_M$ and $\mu$VeLO$_M$ is 1024, we select this width for all our hyperparameter sweeps. Similarly, we use the same depth=3 and training steps=1000 as for the meta-training of $\mu$LO$_M$ and $\mu$VeLO$_M$.

### B.1  TUNING $\mu$ADAM

We tune $\mu$Adam's learning rate ($\eta$) and accumulator coefficients ($\beta_1$, and $\beta_2$). Table 6 reports all hyperparameter values that we swept for each task. Table 7 reports the best-performing hyperparameter values found by selecting the values that achieved the lowest final smoothed training loss on each task.

Table 6: **Hyperparameter sweep values for $\mu$Adam.**

| Hyperparameter | # | Values |
|---|---|---|
| $\eta$ | 32 | $\{10^{-6}, 1.56 \times 10^{-6}, 2.44 \times 10^{-6}, 3.81 \times 10^{-6}, 5.95 \times 10^{-6}, 9.28 \times 10^{-6},$ $1.45 \times 10^{-5}, 2.26 \times 10^{-5}, 3.53 \times 10^{-5}, 5.52 \times 10^{-5}, 8.62 \times 10^{-5},$ $1.35 \times 10^{-4}, 2.10 \times 10^{-4}, 3.28 \times 10^{-4}, 5.12 \times 10^{-4}, 8.00 \times 10^{-4},$ $1.25 \times 10^{-3}, 1.95 \times 10^{-3}, 3.05 \times 10^{-3}, 4.76 \times 10^{-3}, 7.43 \times 10^{-3},$ $1.16 \times 10^{-2}, 1.81 \times 10^{-2}, 2.83 \times 10^{-2}, 4.42 \times 10^{-2}, 6.90 \times 10^{-2},$ $1.08 \times 10^{-1}, 1.68 \times 10^{-1}, 2.63 \times 10^{-1}, 4.10 \times 10^{-1}, 6.40 \times 10^{-1}, 1\}$ |
| $\beta_1$ | 4 | $\{0.85, 0.9, 0.95, 0.99\}$ |
| $\beta_2$ | 4 | $\{0.9, 0.95, 0.99, 0.999\}$ |
| **Total** | 512 | – |

### B.2  TUNING ADAMW

We tune AdamW's learning rate ($\eta$), accumulator coefficients ($\beta_1$, and $\beta_2$), and weight decay ($\lambda$). Table 8 reports all hyperparameter values that we swept for each task. Table 9 reports the best-performing hyperparameter values found by selecting the values that achieved the lowest final smoothed training loss on each task.

Table 7: **Strongest performing hyperparameter values of $\mu$Adam for each task, with and without a schedule.** All optimizers with a schedule use a linear warmup and cosine decay schedule with the minimum learning rate set to $0.1\eta$.

| Task | $\eta$ | $\beta_1$ | $\beta_2$ | GPU Hours |
|---|---|---|---|---|
| $\mathcal{T}^{\text{LM}}_{(3,1024)}$ | 0.1077 | 0.85 | 0.999 | 48 |
| $\mathcal{T}^{\text{ViT}}_{(3,1024)}$ | 0.044173 | 0.85 | 0.999 | 17 |
| IN32$\mathcal{T}^{\text{MLP}}_{(3,1024)}$ | 0.044173 | 0.85 | 0.999 | 9 |
| IN64$\mathcal{T}^{\text{MLP}}_{(3,1024)}$ | 0.028289 | 0.85 | 0.99 | 19 |
| C10$\mathcal{T}^{\text{MLP}}_{(3,1024)}$ | 0.1473 | 0.9 | 0.95 | 4 |
| **Total** | – | – | – | 97 |

Table 8: **Hyperparameter sweep values for AdamW.**

| Hyperparameter | # | Values |
|---|---|---|
| $\eta$ | 14 | $\{0.1,\ 4.92 \times 10^{-2},\ 2.42 \times 10^{-2},\ 1.19 \times 10^{-2},$ $5.88 \times 10^{-3}, 2.89 \times 10^{-3}, 1.43 \times 10^{-3},$ $7.02 \times 10^{-4}, 3.46 \times 10^{-4}, 1.70 \times 10^{-4},$ $8.38 \times 10^{-5}, 4.12 \times 10^{-5}, 2.03 \times 10^{-5}, 1.00 \times 10^{-5}\}$ |
| $\beta_1$ | 3 | $\{0.9,\ 0.95,\ 0.99\}$ |
| $\beta_2$ | 3 | $\{0.95,\ 0.99,\ 0.999\}$ |
| $\lambda$ | 4 | $\{0.1,\ 0.01,\ 0.001,\ 0.0001\}$ |
| **Total** | 504 | – |

Table 9: **Strongest performing hyperparameter values of AdamW for each task, with and without a schedule.** All optimizers with a schedule use a linear warmup and cosine decay schedule with the minimum learning rate set to $0.1\eta$.

| Task | $\eta$ | $\beta_1$ | $\beta_2$ | $\lambda$ | GPU Hours |
|---|---|---|---|---|---|
| $\mathcal{T}^{\text{LM}}_{(3,1024)}$ | $7.02 \times 10^{-4}$ | 0.9 | 0.99 | 0.001 | 48 |
| $\mathcal{T}^{\text{ViT}}_{(3,1024)}$ | $1.70 \times 10^{-4}$ | 0.9 | 0.999 | 0.01 | 18 |
| IN32$\mathcal{T}^{\text{MLP}}_{(3,1024)}$ | $7.02 \times 10^{-4}$ | 0.9 | 0.999 | 0.01 | 9 |
| IN64$\mathcal{T}^{\text{MLP}}_{(3,1024)}$ | $7.02 \times 10^{-4}$ | 0.9 | 0.99 | 0.001 | 20 |
| C10$\mathcal{T}^{\text{MLP}}_{(3,1024)}$ | $2.89 \times 10^{-3}$ | 0.9 | 0.95 | 0.0001 | 4 |
| **Total** | – | – | – | – | 99 |

## C Meta-training with $\mu$LOs

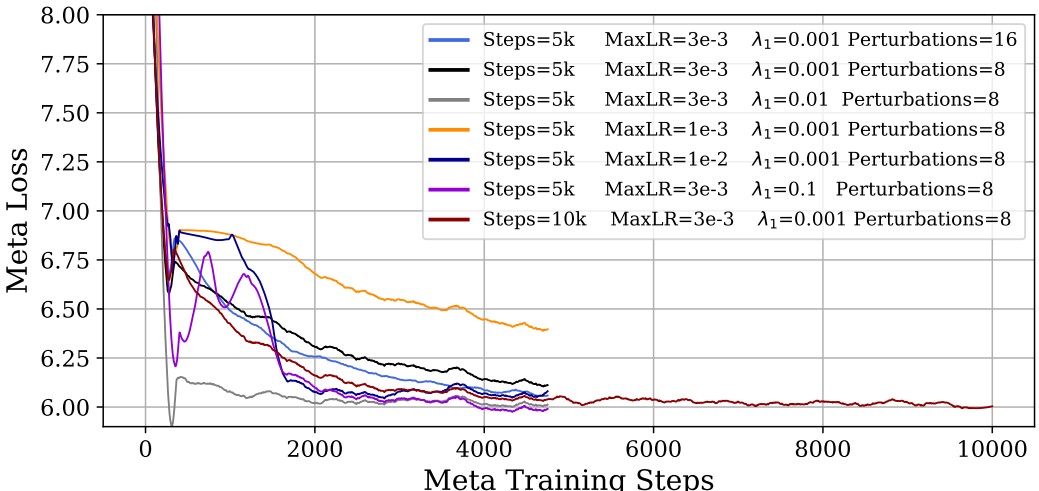

Figure 7: **Ablating Meta-training Hyperparameter for $\mu\text{LO}_S$.** All curves show a single meta-training run. Using AdamW with a linear warmup and cosine annealing schedule, we meta-train $\mu\text{LO}_S$ to train 3-layer width 128 MLPs for classifying $32 \times 32 \times 3$ ImageNet Images. By default, we warmup linearly for 100 steps to a maximum learning rate of $3e - 3$ and anneal the learning rate for 4900 steps to a value of $1e - 3$ with $\lambda_1 = 0.001$ (from Equation 3) and sampling 8 perturbations per step in PESVicol et al. (2021). The above ablation varies the maximum learning rate $\in \{1e - 2, 3e - 3, 1e - 3\}$ (always using 100 steps of warmup and decaying to $0.3 \times$MaxLR), $\lambda_1 \in \{0.001, 0.01, 0.1\}$, the number of steps (5k or 10k), and the number of perturbations (8 or 16). We observe that using all default values except for $\lambda_1 = 0.01$ yields one of the best solutions while being fast to train and stable during meta-training.

**General meta-training setup for small_fc_lopt**  Each small_fc_lopt (Metz et al., 2022a) learned optimizer is meta-trained for 5000 steps of gradient descent using AdamW (Loshchilov & Hutter, 2019) and a linear warmup and cosine annealing schedule. We use PES (Vicol et al., 2021) to estimate meta-gradients with a truncation length of 50 steps and sampling 8 perturbations per task at each step with standard deviation 0.01. For the inner optimization task, we used a maximum unroll length of 1000 iterations; that is, all our learned optimizers see at most 1000 steps of the inner optimization problem during meta-training. Unlike with $\mu$Adam, we do not tune the $\mu$P multipliers when meta-training $\mu\text{LO}_S$ and $\mu\text{LO}_M$, instead, we set them all to 1. All optimizers are meta-trained on a single A6000 GPU. $\mu\text{LO}_S$ and $\text{LO}_S$ take 8 hours each to meta-train, while $\mu\text{LO}_M$ and $\text{LO}_M$ take 103 hours.

**General meta-training setup for VeLO**  Each VeLO (Metz et al., 2022a) learned optimizer is meta-trained for 45000 steps of gradient descent using AdamW (Loshchilov & Hutter, 2019) and a linear warmup and cosine annealing schedule. We using PES (Vicol et al., 2021) to estimate meta-gradients with a truncation length of 20 steps and sampling 8 perturbations per task at each step with standard deviation 0.01. For the inner optimization task, we used a maximum unroll length of 1000 iterations; that is, all our learned optimizers see at most 1000 steps of the inner optimization problem during meta-training. Unlike Yang et al. (2022), we do not tune the $\mu$P multipliers when meta-training $\mu\text{LO}_S$ and $\mu\text{LO}_M$, instead, we set them all to 1. All optimizers are meta-trained on a single A6000 GPU. $\mu\text{VeLO}_M$ and $\text{VeLO}_M$ each take 250 GPU-hours to meta-train.

**Meta-training hyperparameters for small_fc_lopt in $\mu$P**  While there are very few differences between $\mu$LOs and SP LOs, the effective step size for hidden layers is changed (see eq. 3) which could alter the optimal meta-training hyperparameters. Consequently, we conduct an ablation study on hyper-parameters choices for $\mu\text{LO}_S$. Specifically, using AdamW and gradient clipping with a linear warmup and cosine annealing LR schedule, we meta-train $\mu\text{LO}_S$ to train 3-layer width 128

MLPs to classify $32 \times 32 \times 3$ ImageNet Images. By default, we warmup linearly for 100 steps to a maximum learning rate of $3e-3$ and anneal the learning rate for 4900 steps to a value of $1e-3$ with $\lambda_1 = 0.001$ (from Equation 3) and sampling 8 perturbations per step in PESVicol et al. (2021). The above ablation varies the maximum learning rate $\in \{1e-2, 3e-3, 1e-3\}$ (always using 100 steps of warmup and decaying to $0.3 \times$MaxLR), $\lambda_1 \in \{0.001, 0.01, 0.1\}$, the number of steps (5k or 10k), and the number of perturbations (8 or 16). We observe that using all default values except for $\lambda_1 = 0.01$ yields one of the best solutions while being fast to train and stable during meta-training. We, therefore, select these hyperparameters to meta-train $\mu\text{LO}_S$ and $\mu\text{LO}_M$.

**Meta-training hyperparameters for VeLO in $\mu$P**    Unlike for small_fc_lopt, we do not find it necessary to change $\lambda_1$ from its default value of $0.001$. However, we do slightly alter the VeLO update by removing the multiplication by the current parameter norm. This causes problems when initializing tensors to zero, as we do in our experiments.

**$\mu$P at Meta-training time**    It is important to carefully choose meta-training tasks that can effectively be transferred to larger tasks. In (Yang et al., 2022), authors discuss these points and provide two notable guidelines: initialize the output weight matrix to zero (as it will approach zero in the limit) and use a relatively large key size when meta-training transformers. For all our tasks, we initialize the network's final layer to zeros following this guidance. While we do not meta-train on transformers, we suspect that the aforementioned transformer-specific guidelines may be useful for doing so.

# D    EXTENDED RELATED WORK

**Learned optimization.**    While research on learned optimizers (LOs) spans several decades (Schmidhuber, 1992; Thrun & Pratt, 2012; Chen et al., 2022; Amos, 2022), our work is primarily related to the recent meta-learning approaches utilizing efficient per-parameter optimizer architectures of Metz et al. (2022a). Unlike prior work (Andrychowicz et al., 2016; Wichrowska et al., 2017; Chen et al., 2020), which computes meta-gradients (the gradients of the learned optimizer) using backpropagation, Metz et al. (2022a) use Persistent Evolutionary Strategies (PES) (Vicol et al., 2021), a truncated variant of evolutionary strategies (ES) (Buckman et al., 2018; Nesterov & Spokoiny, 2017; Parmas et al., 2018). ES improves meta-training of LOs by having more stable meta-gradient estimates compared to backpropagation through time, especially for longer sequences (i.e. long parameter update unrolls inherent in meta-training) (Metz et al., 2019). PES and most recently ES-Single (Vicol, 2023) are more efficient and accurate variants of ES, among which PES is more well-established in practice making it a favourable approach to meta-training.

**Generalization in LOs.**    One of the critical issues in LOs is generalization in the three main aspects (Chen et al., 2022; Amos, 2022): (1) optimize novel tasks (often referred to as *meta-generalization*); (2) optimize for more iterations than the maximum unroll length used in meta-training; (3) avoid overfitting on the training set. Among these, (3) has been extensively addressed using different approaches, such as meta-training on the validation set objective (Metz et al., 2019), adding extra-regularization terms (Harrison et al., 2022), parameterizing LOs as hyperparameter controllers (Almeida et al., 2021) and introducing flatness-aware regularizations (Yang et al., 2023). The regularization terms (Harrison et al., 2022; Yang et al., 2023) often alleviate issue (2) as a byproduct. However, meta-generalization (1) has remained a more difficult problem.

One approach to tackle this problem is to meta-train LOs on thousands of tasks (Metz et al., 2022b). However, this approach is extremely expensive and does not address the issue in a principled way leading to poor meta-generalization in some cases, e.g. when the optimization task includes much larger networks. Alternatively, Premont-Schwarz et al. (2022) introduced Loss-Guarded L2O (LGL2O) that switches to Adam/SGD if the LO starts to diverge improving meta-generalization. However, this approach needs tuning Adam/SGD and requires additional computation (e.g. for loss check) limiting (or completely diminishing in some cases) the benefits of the LO. In this work, we study aspects (1) and (2) of LO generalization, demonstrating how existing SP LOs generalize poorly across these dimensions and showing how one can apply $\mu$P to learned optimizers to substantially improve generalization in both these aspects.

**Maximal Update Parametrization.** First proposed by Yang & Hu (2021), the Maximal Update Parametrization is the unique stable abc-Parametrization where every layer learns features. The parametrization was derived for adaptive optimizers by Yang & Littwin (2023) and was applied by Yang et al. (2022) to enable zero-shot hyperparameter transfer, constituting the first practical application of the tensor programs series of papers. Earlier works in the *tensor programs series* build the mathematical foundation that led to the discovery of $\mu$P. Yang (2019) shows that many modern neural networks with randomly initialized weights and biases are Gaussian Processes, providing a language, called Netsor, to formalize neural network computations. Yang (2020a) focuses on neural tangent kernels (NTK), proving that as a randomly initialized network's width tends to infinity, its NTK converges to a deterministic limit. Yang (2020b) shows that randomly initialized network's pre-activations become independent of its weights when its width tends to infinity. Most recently, in tensor programs VI, Yang et al. (2024) propose Depth-$\mu$P, a parameterization allowing for hyperparameter transfer in infinitely deep networks. However, Depth-$\mu$P is only valid for residual networks with a block depth of 1, making it unusable for most practical architectures (e.g., transformers, resnets, etc.). For these reasons, we do not study Depth-$\mu$P herein. Building on the latest works studying width $\mu$P (Yang & Littwin, 2023; Yang et al., 2022), in this work, we show that $\mu$P can be extended to the case of learned optimizers and empirically evaluate its benefits in this setting.

# E LIST OF META-TESTING TASKS

Table 10 reports the configuration of different testing tasks used to evaluate our optimizers. We note that we do not augment the ImageNet datasets we use in any way except for normalizing the images. We tokenize LM1B using a sentence piece tokenizer(Kudo & Richardson, 2018) with 32k vocabulary size. All evaluation tasks are run on A6000 48GB or A100 80GB GPUs for 5 random seeds.

# F ADDITIONAL EXPERIMENTS

## F.1 COMPARISON WITH VELO-4000

**Pre-trained VeLO (VeLO-4000).** VeLO (Metz et al., 2022b) is a learned optimizer that was meta-trained on a curriculum of progressively more expensive meta-training tasks for a total of 4000 TPU months. These tasks include but are not limited to image classification with MLPs, ViTs, ConvNets, and ResNets; compression with MLP auto-encoders; generative modeling with VAEs; and language modeling with transformers and recurrent neural networks. During meta-training, VeLO-4000 unrolls inner problems for up to 20k steps ($20\times$ ours); the final model was then fine-tuned on tasks with up to 200k steps of optimization. VeLO-4000, therefore represents a strong but unfair baseline as it is trained on far more data and with far more compute than our main VeLO experiments.

**Is VeLO-4000 a fair baseline?** While we believe the comparison is interesting given the relevance of our results to scaling up LOs, the comparison will unfairly advantage VeLO-4000 as **all tasks in our suite fall within its meta-training distribution** and VeLO-4000 was meta-trained on inner unroll horizons well beyond those we evaluate. Thus, when comparing our LOs to VeLO-4000, it is important to keep in mind that it is an unfair baseline since our learned optimizers meta-trained with only $0.004\%$ of VeLO-4000's compute budget. We included a compute-matched fair baseline, $\text{VeLO}_M$ in the main manuscript.

**Comparison** Figures 8 reports the training curves of different optimizers, including VeLO-4000, on width 8192 and 3072 MLP and transformer language model tasks, respectively. We observe that $\mu\text{LO}_M$ and $\mu\text{VeLO}_M$ (trained with many orders of magnitude less compute) outperforms VeLO-4000 at this large width on the in-distribution tasks, but fall short despite still generalizing well when evaluated far out-of-distribution on a width 3072 language modeling task. We hypothesize that this is likely due to the task being nearly in-distribution for VeLO-4000 meta-training data while being OOD w.r.t. architecture, width, and training steps for $\mu\text{LO}_M$ and $\mu\text{VeLO}_M$. These results overall suggest that $\mu\text{VeLO}_M$ may be more scalable than its non-$\mu$P counterpart, particularly in the large model cases where VeLO-4000 struggled (Metz et al., 2022b).

Table 10: **Meta-testing settings.** We report the optimization tasks we will use to evaluate the LOs of Table 5.

| Identifier | Dataset | Model | Depth | Width | Attn. Heads | FFN Size | Batch Size | Sequence Length |
|---|---|---|---|---|---|---|---|---|
| IN32$\mathcal{T}^{MLP}_{(3,128)}$ | $32 \times 32 \times 3$ ImageNet | MLP | 3 | 128 | – | – | 4096 | – |
| IN32$\mathcal{T}^{MLP}_{(3,256)}$ | $32 \times 32 \times 3$ ImageNet | MLP | 3 | 256 | – | – | 4096 | – |
| IN32$\mathcal{T}^{MLP}_{(3,512)}$ | $32 \times 32 \times 3$ ImageNet | MLP | 3 | 512 | – | – | 4096 | – |
| IN32$\mathcal{T}^{MLP}_{(3,1024)}$ | $32 \times 32 \times 3$ ImageNet | MLP | 3 | 1024 | – | – | 4096 | – |
| IN32$\mathcal{T}^{MLP}_{(3,2048)}$ | $32 \times 32 \times 3$ ImageNet | MLP | 3 | 2048 | – | – | 4096 | – |
| IN32$\mathcal{T}^{MLP}_{(3,4096)}$ | $32 \times 32 \times 3$ ImageNet | MLP | 3 | 4096 | – | – | 4096 | – |
| IN32$\mathcal{T}^{MLP}_{(3,8192)}$ | $32 \times 32 \times 3$ ImageNet | MLP | 3 | 8192 | – | – | 4096 | – |
| IN64$\mathcal{T}^{MLP}_{(3,128)}$ | $64 \times 64 \times 3$ ImageNet | MLP | 3 | 128 | – | – | 4096 | – |
| IN64$\mathcal{T}^{MLP}_{(3,256)}$ | $64 \times 64 \times 3$ ImageNet | MLP | 3 | 256 | – | – | 4096 | – |
| IN64$\mathcal{T}^{MLP}_{(3,512)}$ | $64 \times 64 \times 3$ ImageNet | MLP | 3 | 512 | – | – | 4096 | – |
| IN64$\mathcal{T}^{MLP}_{(3,1024)}$ | $64 \times 64 \times 3$ ImageNet | MLP | 3 | 1024 | – | – | 4096 | – |
| IN64$\mathcal{T}^{MLP}_{(3,2048)}$ | $64 \times 64 \times 3$ ImageNet | MLP | 3 | 2048 | – | – | 4096 | – |
| IN64$\mathcal{T}^{MLP}_{(3,4096)}$ | $64 \times 64 \times 3$ ImageNet | MLP | 3 | 4096 | – | – | 4096 | – |
| C10$\mathcal{T}^{MLP}_{(3,128)}$ | $32 \times 32 \times 3$ Cifar-10 | MLP | 3 | 128 | – | – | 4096 | – |
| C10$\mathcal{T}^{MLP}_{(3,256)}$ | $32 \times 32 \times 3$ Cifar-10 | MLP | 3 | 256 | – | – | 4096 | – |
| C10$\mathcal{T}^{MLP}_{(3,512)}$ | $32 \times 32 \times 3$ Cifar-10 | MLP | 3 | 512 | – | – | 4096 | – |
| $\mathcal{T}^{LM}_{(3,1024)}$ | $32 \times 32 \times 3$ Cifar-10 | MLP | 3 | 1024 | – | – | 4096 | – |
| C10$\mathcal{T}^{MLP}_{(3,2048)}$ | $32 \times 32 \times 3$ Cifar-10 | MLP | 3 | 2048 | – | – | 4096 | – |
| C10$\mathcal{T}^{MLP}_{(3,4096)}$ | $32 \times 32 \times 3$ Cifar-10 | MLP | 3 | 4096 | – | – | 4096 | – |
| C10$\mathcal{T}^{MLP}_{(3,8192)}$ | $32 \times 32 \times 3$ Cifar-10 | MLP | 3 | 8192 | – | – | 4096 | – |
| $\mathcal{T}^{ViT}_{(3,192)}$ | $32 \times 32 \times 3$ ImageNet | ViT | 3 | 192 | 3 | 768 | 1024 | – |
| $\mathcal{T}^{ViT}_{(3,384)}$ | $32 \times 32 \times 3$ ImageNet | ViT | 3 | 384 | 6 | 1536 | 1024 | – |
| $\mathcal{T}^{ViT}_{(3,768)}$ | $32 \times 32 \times 3$ ImageNet | ViT | 3 | 768 | 8 | 3072 | 1024 | – |
| $\mathcal{T}^{ViT}_{(3,1024)}$ | $32 \times 32 \times 3$ ImageNet | ViT | 3 | 1024 | 8 | 4096 | 1024 | – |
| $\mathcal{T}^{ViT}_{(3,2048)}$ | $32 \times 32 \times 3$ ImageNet | ViT | 3 | 2048 | 16 | 8192 | 1024 | – |
| $\mathcal{T}^{ViT}_{(3,3072)}$ | $32 \times 32 \times 3$ ImageNet | ViT | 3 | 3072 | 16 | 12288 | 1024 | – |
| $\mathcal{T}^{LM}_{(3,192)}$ | LM1B, $32k$ Vocab | Transformer LM | 3 | 192 | 3 | 768 | 128 | 64 |
| $\mathcal{T}^{LM}_{(3,384)}$ | LM1B, $32k$ Vocab | Transformer LM | 3 | 384 | 6 | 1536 | 128 | 64 |
| $\mathcal{T}^{LM}_{(3,768)}$ | LM1B, $32k$ Vocab | Transformer LM | 3 | 768 | 8 | 3072 | 128 | 64 |
| $\mathcal{T}^{LM}_{(3,1024)}$ | LM1B, $32k$ Vocab | Transformer LM | 3 | 1024 | 8 | 4096 | 128 | 64 |
| $\mathcal{T}^{LM}_{(3,2048)}$ | LM1B, $32k$ Vocab | Transformer LM | 3 | 2048 | 16 | 8192 | 128 | 64 |
| $\mathcal{T}^{LM}_{(3,3072)}$ | LM1B, $32k$ Vocab | Transformer LM | 3 | 3072 | 16 | 12288 | 128 | 64 |
| $\mathcal{DT}^{MLP}_{(16,1024)}$ | $32 \times 32$ ImageNet | MLP | 16 | 1024 | – | – | 128 | – |
| $\mathcal{DT}^{ViT}_{(16,1024)}$ | $32 \times 32$ ImageNet | ViT | 16 | 1024 | 3 | 4096 | 128 | – |
| $\mathcal{DT}^{LM}_{(16,1024)}$ | LM1B | Transformer LM | 16 | 1024 | 3 | 4096 | 128 | – |

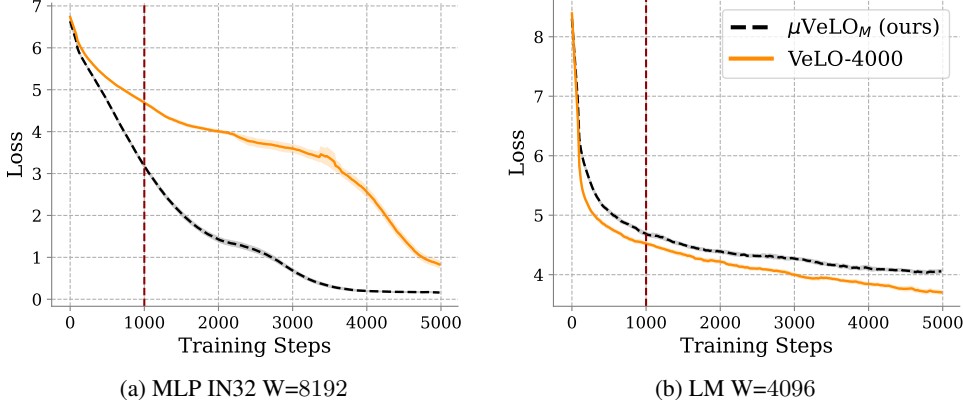

(a) MLP IN32 W=$8192$      (b) LM W=$4096$

Figure 8: **A comparison to VeLO-4000 on the widest tasks.** All optimizers except VeLO are meta-trained or hyperparameter tuned for 1000 inner steps (dotted red line), therefore, any optimization beyond 1000 steps is considered out-of-distribution. We plot average training loss over 5 seeds with standard error bars. We observe that $\mu$LO$_M$ and $\mu$VeLO$_M$ outperform VeLO on the widest in-distribution tasks, but fall short, despite still generalizing well when evaluated far out-of-distribution on a width 3072 language modeling task.

### F.1.1 WHY DO $\mu$LOS IMPROVE GENERALIZATION TO DEPTH AND LONGER TRAINING HORIZONS?

While our goal was to improve the meta-generalization of learned optimizers to unseen wider tasks, in sections 5.2.4 and 5.2.4, we also observed improved meta-generalization to deeper and wider networks. This discovery is entirely empirical as we did not use a parameterization that has depth transfer properties (e.g. $\mu$Depth (Yang et al., 2024)). With Figure 9 as evidence, we hypothesize that the reason for improved transfer to deeper models and longer training is $\mu$LOs' ability to maintain stable logits in the optimizee throughout training in contrast to SP LOs. For instance, in subfigure (a), we observe that the first layer pre-activations of depth 8 and depth 16 MLPs trained with $\text{LO}_M$ grow rapidly at the beginning of training, while those of deeper MLPs optimized by $\mu\text{LO}_M$ vary similarly to the depth-3 MLP (same depth as meta-training). In subfigure (b), we observe a similar but less drastic change in logit L1 norm as training progresses. While the L1 norm of the MLP trained by $\mu\text{LO}_M$ consistently grows at a stable rate throughout training, for $\text{LO}_M$ the MLP's logits undergo a change in slope after 8000 steps of training and a near discontinuity at 13000 steps. With the evidence we have so far, it is not possible to be certain whether the observed activation stability is the cause of the improved generalization or merely a symptom of it. That being said, these results can still help inform on favorable properties for the generalization of LOs.

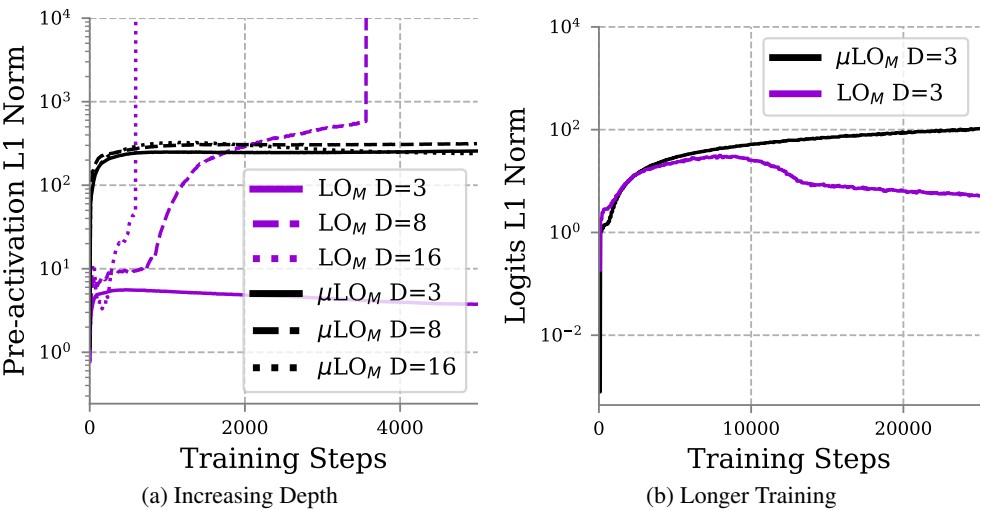

Figure 9: **Activation stability for deeper and longer training.** Each curve reports the five-seed average L1 norm of first-layer pre-activation and logits for (a) and (b), respectively.

### F.1.2 EVALUATING THE PERFORMANCE OF $\mu\text{LO}_M$ ON RESNET TASKS

In this section, we compare the meta-generalization performance of $\mu\text{LO}_M$ to $\text{LO}_M$ on ResNet tasks. Figure 10 reports train and test loss during training, while Table 11 reports the final train and test loss across different width ResNet tasks for each optimizer. We observe that each width $\mu\text{LO}_M$ outperforms $\text{LO}_M$ and that the final evaluation loss for each closely tracks the training loss.

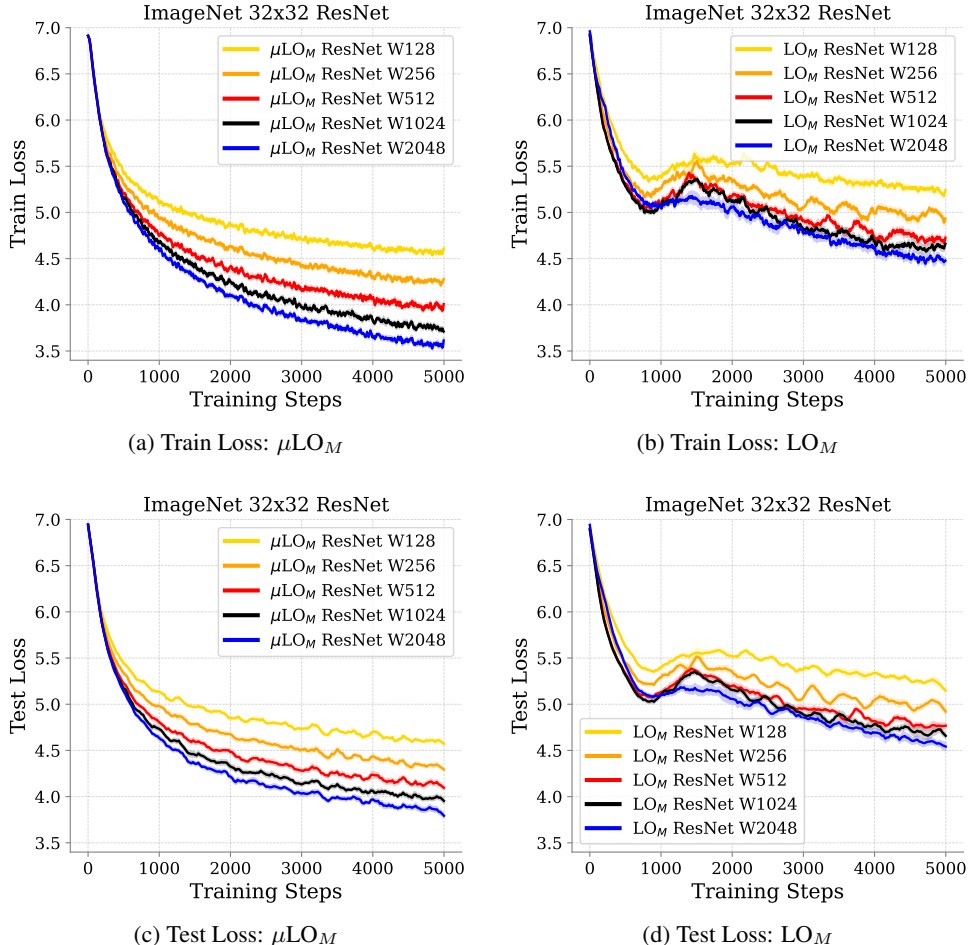

Figure 10: **Train and Test loss across width on a ResNet task.** Each curve reports the five-seed average. We compare the meta-generalization of $\mu\text{LO}_M$ to $\text{LO}_M$ when optimizing ResNet tasks. We find that $\mu\text{LO}_M$ generalizes significantly better.

Table 11: **Final train and test losses on ResNet tasks.** Each value reports the five-seed average and is rounded to two decimals.

| Optimizer | Width | Final Train Loss | Final Test Loss |
|---|---|---|---|
| $\mu\text{LO}_M$ | 128 | $4.56 \pm 0.03$ | $4.58 \pm 0.03$ |
| $\mu\text{LO}_M$ | 256 | $4.25 \pm 0.02$ | $4.32 \pm 0.03$ |
| $\mu\text{LO}_M$ | 512 | $3.95 \pm 0.02$ | $4.11 \pm 0.02$ |
| $\mu\text{LO}_M$ | 1024 | $3.72 \pm 0.03$ | $3.98 \pm 0.05$ |
| $\mu\text{LO}_M$ | 2048 | $3.56 \pm 0.02$ | $3.83 \pm 0.03$ |
| $\text{LO}_M$ | 128 | $5.20 \pm 0.01$ | $5.18 \pm 0.02$ |
| $\text{LO}_M$ | 256 | $4.92 \pm 0.04$ | $4.93 \pm 0.03$ |
| $\text{LO}_M$ | 512 | $4.70 \pm 0.02$ | $4.76 \pm 0.02$ |
| $\text{LO}_M$ | 1024 | $4.63 \pm 0.02$ | $4.70 \pm 0.05$ |
| $\text{LO}_M$ | 2048 | $4.47 \pm 0.03$ | $4.59 \pm 0.05$ |

# G  Coordinate evolution of MLP layers in $\mu$P for Adam and Learned Optimizers

The following section presents the continuation of our experiments comparing pre-activation growth during training for SP LOs and $\mu$LOs with different meta-training recipes, SP adam, and $\mu$Adam.

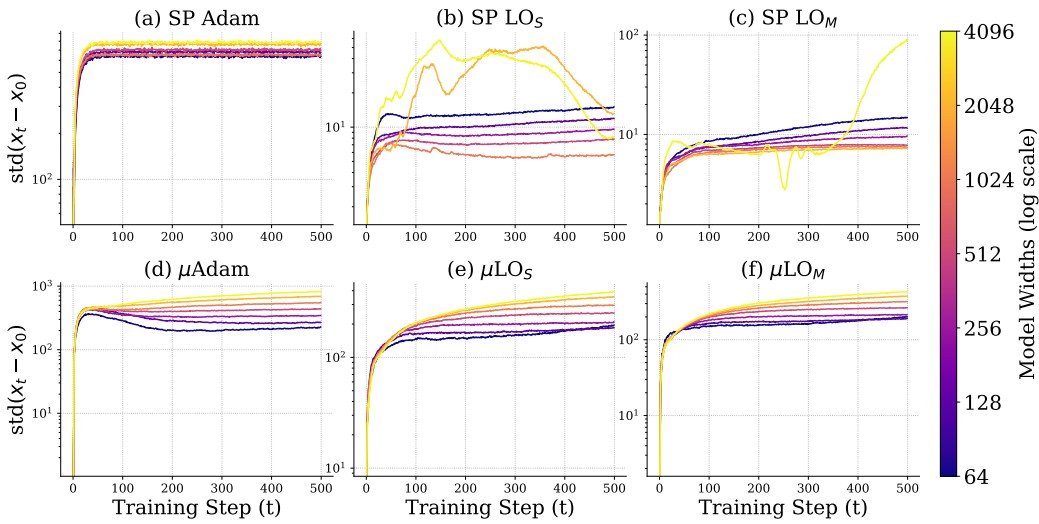

Figure 11: **Layer 0 pre-activations behave harmoniously in $\mu$P for LOs and Adam alike.** We report the evolution of coordinate-wise standard deviation between the difference of initial and current second-layer pre-activations. We observe that all models parameterized in $\mu$P enjoy stable coordinates across widths, while the pre-activations of larger-width models in SP blow up after a number of training steps. All plots report these metrics for the first 500 steps of a single training run.

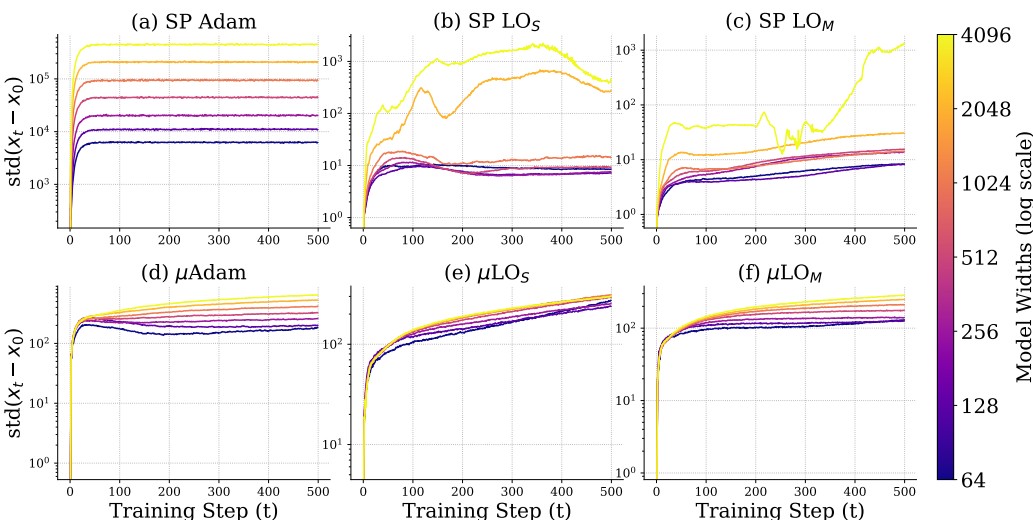

Figure 12: **Layer 1 pre-activations behave harmoniously in $\mu$P for LOs and Adam alike.** We report the evolution of coordinate-wise standard deviation between the difference of initial and current second-layer pre-activations. We observe that all models parameterized in $\mu$P enjoy stable coordinates across widths, while the pre-activations of larger-width models in SP blow up after a number of training steps. All plots report these metrics for the first 500 steps of a single training run.

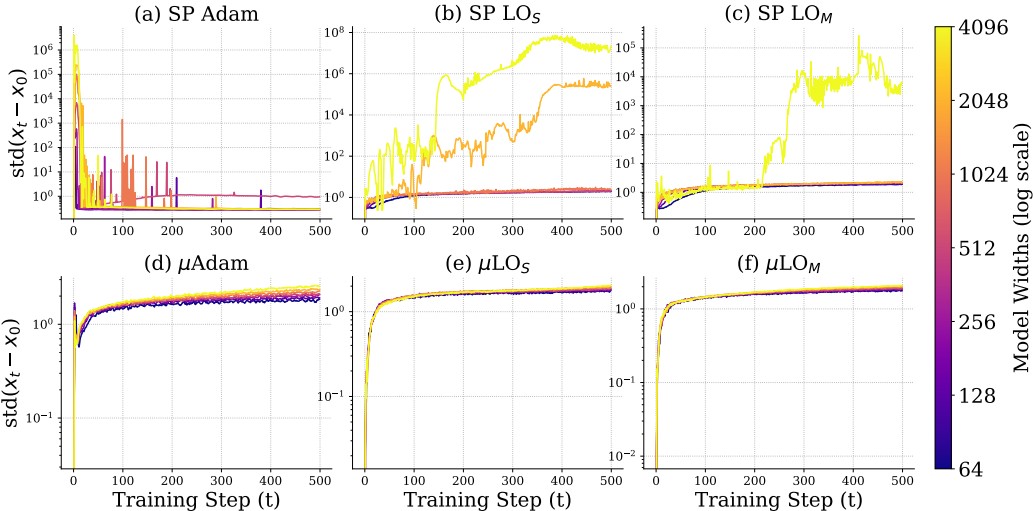

Figure 13: **Logits behave harmoniously in $\mu$P for LOs and Adam alike.** We report the evolution of coordinate-wise standard deviation between the difference of initial and current second-layer pre-activations. We observe that all models parameterized in $\mu$P enjoy stable logits across widths, while the pre-activations of larger-width models in SP blow up after a number of training steps. All plots report these metrics for the first $500$ steps of a single training run.