# OpenReview forum: "$\mu$LO: Compute-Efficient Meta-Generalization of Learned Optimizers"
_ICLR.cc/2026/Conference — ICLR 2026 Poster_

### Official Review · Reviewer_oqd9 · 2025-10-30

**Soundness:** 3
**Presentation:** 3
**Contribution:** 3
**Rating:** 6
**Confidence:** 4

**Summary:**

This paper introduces \mu-parameterized Learned Optimizers to address the meta-generalization limitations of existing learned optimizers when optimizing neural networks wider than those encountered during meta-training.

Authors derive the Maximal Update Parametrization  for some architectures and propose a straightforward meta-training recipe. They find that proposed \mu-P parameterization significantly improves meta-generalization to wider, deeper, and longer unseen tasks compared to Standard Parameterization.

**Strengths:**

The paper has following strengths in my opinion:
- tackles the fundamental challenge of meta-generalization in learned optimizers, which is a major hurdle for their practical adoption.
- derivation of µP for LOs provides a solid theoretical basis for the proposed approach
- results show substantial improvements in meta-generalization across various axes (width, depth, training horizons) compared to  Standard Parametrization
- authors emphasize that the benefits of µLOs come at zero extra computational cost during meta-training compared to SP LOs, which is a crucial practical advantage
- paper provides a clear description of the meta-training recipe, evaluation setup, and baselines, making the work reproducible
- good suite of evaluation tasks and architectures

**Weaknesses:**

My main concerns are following:
- While the improved generalization to deeper networks and longer training horizons is a strong empirical result, the explanations for these benefits are currently speculative e.g. mentions of pre-activation stability. Wondering if this could be made more precise.

**Questions:**

I also have some qns for the authors:
- Given that µLOs exhibit improved generalization to deeper networks and longer training horizons, and these improvements are currently hypothesized to be due to activation stability, do the authors plan to conduct further theoretical analysis to formalize these observations?
- The µLOs were meta-trained only on MLP tasks. How do the authors anticipate the performance and meta-training cost of µLOs would change if they were meta-trained on a more diverse set of architectures, including ViTs and Transformers?
- While the paper acknowledges the absence of an oracle SP AdamW baseline, could the authors discuss any ongoing efforts or future plans to conduct such a computationally intensive comparison to provide an even stronger benchmark?

---

> ### Author Response · Authors · 2025-11-20
>
> We would like to thank oqd9 for reviewing our paper. We are pleased that the reviewer believes our method shows substantial improvements in meta-generalization across many axes, that our paper is clear enough to allow for reproducing the results, and that we selected a good suite of evaluation tasks.
>
> ---
> ## Answers to Weaknesses
> **Better understanding benefits for deeper networks and longer training horizons** We would like to emphasize that we do provide further analysis regarding improved meta-generalization to depth and width in section F.1.1 of the appendix. Specifically, we show that increasing depth and training durations systematically cause pre-activations to shift in SP models, while they remain stable for $\mu$LOs. However, as mentioned by the reviewer, our analysis remains speculative as our evidence only establishes a correlation between improved pre-activation stability and better meta-generalization. Unfortunately, it is difficult to make this more precise, since learned optimizers themselves are neural networks. Therefore, our ability to establish cause-and-effect relationships regarding training outcomes of $\mu$LOs is restricted by our field's already limited ability to interpret the decisions of neural networks. Improving this is beyond the scope of our paper.
>
> ---
> ## Answers to Reviewer questions
>
> **Further theory for depth improvements** Given our primary focus on width generalization in the paper, we do not intend to conduct further theoretical analysis for depth or unroll generalization. As stated on lines 450-451 our findings for depth and longer training are purely empirical.
>
> **Expected Performance as meta-training tasks are scaled** In short, we expect that increasing the number of tasks within the meta-training distribution will result in performance improvements across all tasks within this distribution and their larger-width counterparts. Our expectation follows from existing work [4.1], which finds that scaling the number of tasks included in the meta-training distribution results in a lower loss across all in-distribution tasks. Therefore, if we were to include more tasks in the meta-training distribution of $\mu$LOs, we would expect the resulting optimizer to perform better overall across these tasks, and *crucially, we would also expect it to perform better across wider counterparts of the same tasks*.
>
> **Expected meta-training cost as meta-training tasks are scaled** We expect meta-training costs to increase as the meta-training distribution is scaled to larger tasks such as ViTs and LMs. However, we believe that $\mu$LOs will be important for keeping these costs down, even as tasks are scaled up, since $\mu$LOs do not require training on larger-width versions of the tasks, unlike existing work using standard parameterization.
>
> **Oracle SP AdamW baseline** We would like to highlight that $\mu$Adam represents a very strong baseline here, which is comparable to an oracle. $\mu$Adam is tuned using an extensive grid search across over 500 hyperparameter configurations (see Table 6) on a small-width version of each task. Therefore, since $\mu$Adam  uses $\mu$-parameterization[4.2], its optimal hyperparameters transfer from small-width to large-width tasks. This means that, even when applied to larger tasks,  $\mu$Adam’s hyperparameters are optimal in the main experiments of our paper (the width scaling study of Figure 5 and Table 1). Moreover, since $\mu$Adam uses hyperparameters tuned on each task, *the baseline is advantaged relative to the learned optimizers in our study, which are only meta-trained on MLP tasks*.  We believe that the $\mu$Adam is a sufficiently strong baseline for contextualizing the performance of our method. That being said, we now provide the final loss for an AdamW tuning task: width=1024 MLP at 1000 steps. AdamW reaches: 5.07 compared to 4.89 for $\mu$LO$_M$. In the camera-ready version, we plan to add these results for other tasks in the paper.
>
>
> ---
> **References**
>
> [4.1][Tasks, stability, architecture, and compute: Training more effective learned optimizers, and using them to train themselves]
>
> [4.2][Tensor Programs V: Tuning Large Neural Networks via Zero-Shot Hyperparameter Transfer, Neurips 2021]

---

> ### Author Response · Authors · 2025-11-27
>
> Dear Reviewer oqd9,
>
> Thank you for your valuable feedback. We have addressed all your questions and concerns in the revised paper and our detailed reply above.
>
> If our reply has addressed your outstanding concerns, we ask you to please consider raising your score.
>
> Best regards,
>
> -The Authors

---

### Official Review · Reviewer_P3eP · 2025-10-31

**Soundness:** 3
**Presentation:** 3
**Contribution:** 3
**Rating:** 6
**Confidence:** 2

**Summary:**

This paper addresses the challenge of learned optimizers (LOs) struggling to optimize unseen tasks, particularly when dealing with networks larger than those encountered during meta-training. The authors introduce the Maximal Update Parametrization (P) for two state-of-the-art LO architectures and propose a new meta-training recipe for parameterized LOs. Empirical results show that LOs trained with this recipe significantly improve meta-generalization to larger unseen tasks, outperforming standard parametrization (SP) LOs within the same compute budget. Additionally, the paper observes enhanced meta-generalization for deeper networks and surprisingly effective generalization for longer training horizons compared to SP LOs.

**Strengths:**

1. This paper introduces the Maximal Update Parametrization to address the meta-generalization problem in learned optimizers. The idea is novel and interesting.

2. The experimental results are thorough and effectively demonstrate the method’s validity.

**Weaknesses:**

1. I recommend adding experiments with convolutional neural networks (CNNs). Although this limitation is mentioned, I believe CNNs are currently mainstream in neural network research, and testing the method on them would further strengthen the validity of the findings.

I am not very familiar with this field, so I will rely on the feedback from other reviewers for the final score.

**Questions:**

N/A

---

> ### Author Response · Authors · 2025-11-20
>
> We would like to thank P3eP for reviewing our paper. We are pleased that the reviewer believes our paper is ”novel and interesting” and that our experiments are thorough.
>
> **Adding results on CNN** We thank the reviewer for their suggestion. We have now added additional ImageNet experiments on ResNets in Section F.1.2 of the appendix (lines 1497-1565). We also provide the results of our comparison in-line here in the following table. We observe that $\mu$LO$_M$ outperforms LO$_M$ on ResNet tasks across all widths we study.
>
> | Optimizer  | Width | Final Train Loss        | Final Test Loss         |
> |------------|-------|-------------------|-------------------|
> | $\mu$LO$_M$ | 128   | 4.56 ± 0.03        | 4.58 ± 0.03        |
> | $\mu$LO$_M$ | 256   | 4.25 ± 0.02        | 4.32 ± 0.03        |
> | $\mu$LO$_M$ | 512   | 3.95 ± 0.02        | 4.11 ± 0.02        |
> | $\mu$LO$_M$ | 1024  | 3.72 ± 0.03        | 3.98 ± 0.05        |
> | $\mu$LO$_M$ | 2048  | 3.56 ± 0.02        | 3.83 ± 0.03        |
> | LO$_M$     | 128   | 5.20 ± 0.01        | 5.18 ± 0.02        |
> | LO$_M$     | 256   | 4.92 ± 0.04        | 4.93 ± 0.03        |
> | LO$_M$     | 512   | 4.70 ± 0.02        | 4.76 ± 0.02        |
> | LO$_M$     | 1024  | 4.63 ± 0.02        | 4.70 ± 0.05        |
> | LO$_M$     | 2048  | 4.47 ± 0.03        | 4.59 ± 0.05        |

---

> ### Author Response · Authors · 2025-11-27
>
> Dear Reviewer P3eP,
>
> Thank you for your valuable feedback. We have addressed all your questions and concerns in the revised paper and our detailed reply above.
>
> If our reply has addressed your outstanding concerns, we ask you to please consider raising your score.
>
> Best regards,
>
> -The Authors

---

### Official Review · Reviewer_PtQP · 2025-11-01

**Soundness:** 4
**Presentation:** 4
**Contribution:** 3
**Rating:** 6
**Confidence:** 3

**Summary:**

The paper derives muP for the small_fc_lopt and VeLO learned optimizer architectures. The authors then test the resulting metageneralization, meta-training on ImageNet classification with MLPs and applying the optimizers to other instances of that as well as using transformers for that and language modelling. They empirically find that this achieves hyperparameter transfer not only for width, but also to a significant degree for depth and training duration, outperforming other learned optimizers and also competing with task-tuned standard/hand-designed ones like AdamW and μAdam. There is also a short subsection on pre-activation stability, and similar investigations in the appendix.

**Strengths:**

While I think applying muP to this domain was inevitable, the authors make a good case that the failure to meta-generalize is the major blocker for LOs, and that they make substantial improvements there.

I thought the experiments were reasonable, namely the baselines, and the paper was clear throughout, including the maths (though I haven't gone line by line in the proofs).

**Weaknesses:**

I think the findings of ["Scaling Exponents Across Parameterizations and Optimizers"](https://arxiv.org/abs/2407.05872) should've made an appearance somewhere, since they indicate that standard parametrization can also achieve hyperparameter transfer. They also show that (in larger problem instances than here) that epsilon should be tuned in Adam.

Also, since depth scaling is mentioned (albeit as a bonus), I think the related works and perhaps some experiments would ideally address more heuristic methods for scaling with depth (and width, or rollout length, if the LO community has investigated this at all). In NeurIPS 2025 (i.e., contemporary, ignoring arXiv) work, [ComputeP](https://arxiv.org/abs/2505.01618) is quite relevant

Minor fix:
* §B.1 "When using a schedule, we always use linear warmup and cosine annealing with" just cuts off

**Questions:**

How much compute went into the hyperparameter sweeps for the baseline optimizers? The values swept looked reasonable, but I think it'd be useful context to compare with the meta-training figures, to see how much the LOs need to be used to make up for that (mainly thinking in seemingly underparametrized settings where that may be the right frame, like language modelling).

---

> ### Author Response · Authors · 2025-11-20
>
> We would like to thank PtQP for reviewing our paper. We are pleased that the reviewer believes our paper is well written, that we do a good job motivating the work, and that our method makes substantial improvements to learned optimizer (LO) meta-generalization
>
> **Regarding scaling exponents across parameterizations** We agree that [2.2] is relevant to our work. As such, we have added a discussion of this in the related work section (lines 170-174), included a caveat about epsilon underflow in our method section (lines 189-191), and explained that LO meta-generalization under different parameterizations is an important direction for future work (lines 496-497).
>
> **Regarding CompleteP** At the time of writing, CompleteP was concurrent unpublished work. As such, we were unable to include it in the manuscript at that time. Given the relevance of CompleteP to our work, we have now discussed it in our related work (lines 165-166) and future work sections (lines 497-499).
>
> **Minor fix** Thank you, we have fixed this in the updated version.
>
> **Hyperparameter tuning compute** For each task, we perform a grid search over 500 different hyperparameter values for each hand-designed baseline (see Section B). The number of GPU Hours required for each task are reported in Tables 7 and 9 of the appendix. Across all task types in our evaluation suite, the total is approximately 100 GPU hours for each hand-designed optimizer.
>
> ---
> **References**
>
> [2.1][Don't be lazy: CompleteP enables compute-efficient deep transformers, NeurIPS 2025]
>
> [2.2][scaling exponents across parameterizations and optimizers, ICML 2024]
>
> [2.3][Tensor Programs V: Tuning Large Neural Networks via Zero-Shot Hyperparameter Transfer, NeurIPS 2021]

---

> ### Author Response · Authors · 2025-11-27
>
> Dear Reviewer PtQP,
>
> Thank you for your valuable feedback. We have addressed all your questions and concerns in the revised paper and our detailed reply above.
>
> If our reply has addressed your outstanding concerns, we ask you to please consider raising your score.
>
> Best regards,
>
> -The Authors

---

### Official Review · Reviewer_tB8g · 2025-11-10

**Soundness:** 2
**Presentation:** 2
**Contribution:** 2
**Rating:** 2
**Confidence:** 3

**Summary:**

The authors of this submission investigate the generalization ability of learned optimizers in zero-shot-like scenarios, where an optimizer trained on one architecture can be directly applied to unseen architectures. To achieve this, models are initialized following the $\mu$P (Maximal Update Parameterization) principle. The authors leverage their hyperparameter reuse property, which allows optimal parameters discovered on one architecture to be transferred to architectures with the same depth but different widths via a well-defined mapping. Under this framework, learned optimizers are treated as a form of hyperparameter, enabling transfer to new architectures. The proposed approach is evaluated across various neural network training tasks and demonstrates promising results.

**Strengths:**

1. The motivation is interesting and well-grounded. The idea of viewing optimizers as hyperparameters that can generalize across $\mu$P-initialized networks of varying widths is both novel and conceptually appealing.

2. The learned optimizers show encouraging results in zero-shot-like transfer scenarios.

**Weaknesses:**

1. Figure 5 indicates the meta overfitting with the trained optimiser performing well on the meta-train tasks while failing to generalise well on the unseen tasks.

2. From all the experiments, only the learning curve of training stages is illustrated, with the question of whether the learned optimizer leads to advanced generalization ability not answered.

3. Limited novelty: the Mup parameterization proposed in this submission is very close to a direct application of the original $\mu$P paper. In addition, the propositions in submission are similar to the theoretical results from the $\mu$P papers.

**Questions:**

1. Figure 5 suggests meta-overfitting: the learned optimizer performs well on meta-training tasks but fails to generalize effectively to unseen tasks.

2. Across the experiments, only training curves are presented. It remains unclear whether the learned optimizer contributes to improved generalization performance beyond faster or more stable training.

3. The level of novelty appears limited. The proposed $\mu$P parameterization closely follows the original $\mu$P paper, and several propositions in the submission echo theoretical results already established in prior work.

**Details Of Ethics Concerns:**

1. Did the authors adapt VeLO and LO to the experimental tasks? More specifically, were VeLO and LO meta-trained and evaluated under the same protocol used for the proposed learned optimizer?

2. The ViT model was trained using Adam. Is there a reason this optimizer was not included in the comparison? Similarly, why was SGD-M omitted for MLP training?

3. The batch sizes used in the experiments seem unusually large. Many experiments use batch sizes of 1024 or 4096. Could the authors clarify the rationale behind using such large batch sizes?

---

> ### Author Response · Authors · 2025-11-20
>
> We would like to thank tB8g for taking the time to review our paper. We are pleased that the reviewer believes that our method is “both novel and conceptually appealing.” and that they believe our results are encouraging.
>
> ---
> ## Reply to Weaknesses
> **Meta Overfitting in Figure 5** We respectfully disagree with the reviewer and ask to re-evaluate this assessment. On lines 431-452, we explain how Figure 5 shows our proposed optimizers ($\mu$LO$_M$ and $\mu$VeLO$_M$) generalize to new tasks, while the learned optimizer baselines LO$_M$ and VeLO$_M$ diverge. As highlighted in the text, Figure 5 shows that our optimizers (black color) meta-generalize (e.g., generalize to unseen tasks) significantly better than compute-matched learned optimizers from existing work (e.g., purple curves). Please let us know if there is anything unclear.
>
> **Test Loss** We have added an evaluation of optimizee generalization in Tables 1 and 2 below, showing that the final train loss tracks closely to the final test loss for $\mu$LOs. We agree with the review that it is important to evaluate the generalization of optimizees. However, we would like to emphasize that learned optimizer meta-generalization, **the focus of our study**, does not require evaluating optimizee generalization. We further elaborate on this point in our response below.
>
> **Table 1. Test and Train loss for ImageNet 32x32 ViT**
> | Optimizer     | Width | Final Train Loss      | Final Test Loss       |
> |---------------|-------|------------------------|------------------------|
> | $\mu$LO$_M$   | 1024  | 5.0973 ± 0.0412        | 5.2218 ± 0.0329        |
> | $\mu$LO$_M$   | 2048  | 4.984 ± 0.0376         | 5.1497 ± 0.0332        |
> | LO$_M$        | 1024  | 6.4514 ± 0.0108        | 6.4843 ± 0.0177        |
> | LO$_M$        | 2048  | 6.568 ± 0.0166         | 6.5869 ± 0.0106        |
>
>
> **Table 2. Test and Train loss  Language Modelling on LM1B**
> | Optimizer     | Width | Final Train Loss      | Final Test Loss       |
> |---------------|-------|------------------------|------------------------|
> | $\mu$LO$_M$   | 1024  | 4.0324 ± 0.0195        | 4.0617 ± 0.0257        |
> | $\mu$LO$_M$   | 2048  | 4.0101 ± 0.0446        | 4.0264 ± 0.0344        |
> | LO$_M$        | 1024  | 5.1261 ± 0.026         | 5.1162 ± 0.0504        |
> | LO$_M$        | 2048  | 9.741 ± 0.9213         | 9.9651 ± 0.3396        |
>
> **Meta generalization v.s. Optimizee generalization are distinct**  We would like to clarify this subtle point. Meta-generalization can be evaluated regardless of whether the meta-objective targets training or validation loss. Since our meta-objective is the per-timestep training loss of the underlying optimization tasks, we report training loss when evaluating our models. Since meta-generalization evaluates the ability of the learned optimizer to successfully optimize unseen tasks, *not the ability of the optimizee to generalize to unseen data*, our choice to report training loss does not prevent us from faithfully evaluating meta-generalization in a scientific manner. **Why study training loss?** The main focus of our paper is to study the meta-generalization of $\mu$LOs applied to larger-width networks. Therefore, similar to prior published work [1.4], we choose to focus study training loss for simplicity. Studying the generalization of the optimizer to unseen data is an important orthogonal direction that has been explored multiple times in previous works, as mentioned in L134-138. Specifically, it is well known that the following strategies can improve optimizee generalization: targeting validation loss during meta-training [1.1], using weight decay [1.2],  or using flatness-aware regularization [1.3]. This being said, we have now included an evaluation of optimizee generalization in the tables above.

---

> ### Author Response · Authors · 2025-11-20
>
> **Limited novelty**  While we recognize that that the Maximal update parameterization we obtain for $\mu$LOs is equivalent to that of Adam from [1.6], we contend that our work still represents a significant and novel contribution for the following reasons: (1) we are the first to establish that there exists a synergistic connection between meta-generalization in learned optimizers (LOs) and hyperparameter transfer via Maximal Update Parametrizations (MuP) and (2) we are the first work to identify the correct  $\mu$-parameterization for learned optimizers.
>
>
> The connection we establish between meta-generalization in learned optimizers (LOs) and hyperparameter transfer via Maximal Update Parametrizations (MuP) is synergistic, unlocking capabilities beyond what either approach offers independently. The **meta-generalization abilities acquired during meta-training allow LOs to generalize to unseen tasks and architectures**, while **MuP enables generalization to wider tasks of the same architecture** by transferring hyperparameters from smaller to larger networks without re-tuning. By uniting these strengths, our μLOs are the first learned optimizers capable of generalizing to large, unseen tasks.
>
> Deriving the appropriate MuP for LOs is a non-trivial and novel contribution that establishes a new understanding of learned optimizer architectures for the field. Depending on the optimizer’s update, a different maximal update parameterization may be required. Our work provides the first-ever derivation (Section 4 and Section A of the appendix) for learned optimizers, which spans two popular architectures, including VeLO, the SOTA LO architecture [1.5].
>
> ## Reply to Questions
>
>
> **Were baselines (VeLO and LO) trained on the same tasks as muLO and muVeLO?** Yes, all learned optimizers are trained on the same set of 3 MLP imagenet classification tasks. Therefore, they are FLOP and task-matched, meaning that they use the same meta-training budget and meta-training tasks. Table 5 of the appendix reports this information in detail. This also means that in our experiments, the learned optimizers are disadvantaged relative to hand-designed optimizers that are tuned specifically on each task.
>
> **Adam and SGDM not included** We would like to highlight that $\mu$Adam is an equivalent baseline to Adam, whose hyperparameters transfer to larger models [1.6]. As such, adding an additional Adam baseline would be redundant. In addition to $\mu$Adam,  we chose to include the stronger AdamW [1.8] baseline. We chose not to include SGDM so as to keep the focus of our graphs on adaptive optimizers frequently used for practical workloads. However, in light of the rebuttal questions, we have added a comparison in the table below for the 8192-dimensional MLP classification task. We observe that $\mu$LO$_M$ performs best, with SGDM performing similarly to $\mu$Adam.
>
> Table 3. Loss at 1000 and 5000 steps for $\mu$LO$\_M$, SGDM, and $\mu$Adam on the 8192-dimensional MLP classification task.
> | Steps | $\mu$LO\_M | SGDM | $\mu$Adam |
> |-------|------------|------|-----------|
> | 1000   | 3.98       | 4.43 | 4.58     |
> | 5000  | 0.27       | 0.50 | 0.87      |
>
>
>
> **Rationale for using large batch sizes** Given our paper’s goal of improving the generalization of learned optimizers to wider tasks, we decided to use larger batch sizes, which are frequently used in large-scale training [1.7,1.9]. For example, [1.7] trains ViTs with 4096 batch size.
>
>
>
> ---
> **Local References**
>
> [1.1] [Understanding and correcting pathologies in the training of learned optimizers, ICML 2019]
>
> [1.2] [A Closer Look at Learned Optimization: Stability, Robustness, and Inductive Biases, Neurips 2022]
>
> [1.3]  [Learning to Generalize Provably in Learning to Optimize, AISTATS 2023]
>
> [1.4] [Practical Tradeoffs Between Memory, Compute, and Performance in Learned Optimizers, CoLLAs 2022]
>
> [1.5] [VeLO: Training Versatile Learned Optimizers by Scaling Up]
>
> [1.6] [Tensor Programs V: Tuning Large Neural Networks via Zero-Shot Hyperparameter Transfer, Neurips 2021]
>
> [1.7][An Image is Worth 16x16 Words: Transformers for Image Recognition at Scale; ICLR2021]
>
> [1.8][Decoupled Weight Decay Regularization; ICLR 2019]
>
> [1.9][The Llama 3 Herd of Models]

---

> ### Author Response · Authors · 2025-11-27
>
> Dear Reviewer tB8g,
>
> Thank you for your valuable feedback. We have addressed all your questions and concerns in the revised paper and our detailed reply above.
>
> If our reply has addressed your outstanding concerns, we ask you to please consider raising your score.
>
> Best regards,
>
> -The Authors

---

> > ### Comment · Reviewer_tB8g · 2025-11-27
> >
> > Thanks for your detailed response, and all my concerns are addressed. I will raise my score.

---

### Author Response · Authors · 2025-11-20
**Paper update 1**

Dear reviewers,

We would like to thank you all for taking the time to review our paper.

In response to your thoughtful feedback, we have made the following changes to the manuscript (highlighted in blue):
- We have added experiments on ResNet tasks in section F.1.2 of the appendix.
- We have included additional references in the related work section and have made related changes to the text.
- We have added a paragraph to the conclusion for future work.

-The Authors

---

### Comment · Area_Chair_oQy7 · 2025-11-27

Dear reviewers,

The authors have provided detailed responses to your reviews. I would appreciate if you could let both me and the authors know how these responses impact your assessment of the paper.

Best,

AC

---

### Author Response · Authors · 2025-12-01
**README for new AC (summary of weaknesses and responses)**

# README for new AC (summary of weaknesses and responses)

&nbsp;

**TLDR:** Reviewer tB8g (lowest 2/10 score) stated, **“all my concerns are addressed. I will raise my score.”** during the discussion phase. Other reviewers (6/10,6/10,6/10) did not get a chance to reply to our comments. We summarize our replies to their weaknesses below.

&nbsp;

### **tB8g** | score: 2/10  confidence: 3/5 | Discussion Reply: “I will raise my score.”
---
- **tB8g weaknesses:** (a) meta-overfitting in figure 5, (b) test loss was not sufficiently provided, and (c) limited novelty.

- **Authors’ reply to tB8g:** We (a) highlight why Figure 5 actually shows that our optimizer actually shows strong meta-generalization (not meta-overfitting), (b) we provide test loss for the ViT and language modelling tasks in our study, showing it tracks closely to train loss, and (c) we explained why our work represents a significant and novel contribution.

- **tB8g reply to Authors:** In their reply, tB8g states: **“Thanks for your detailed response, and all my concerns are addressed. I will raise my score.”**


&nbsp;

### **PtQP** |  score: 6/10  confidence: 3/5 | Discussion reply: N/A
---

- **PtQP weaknesses:** Missing references [1] and [2].

- **Authors’ reply to PtQP:** We have incorporated both missing references into the related work and future work sections.

&nbsp;

### **P3eP** |  score: 6/10  confidence: 2/5 | Discussion reply: N/A
---

- **P3eP weaknesses:** Missing experiments on CNNs.

- **Authors’ reply to P3eP:** We added new ResNet experiments to our paper, demonstrating that our $\mu$LOs also show improved meta-generalization to CNN tasks.


&nbsp;

### **oqd9** |  score: 6/10  confidence: 4/5 | Discussion reply: N/A
---

- **oqd9 weaknesses:** The reviewer asks for a better explanation of our empirical observation that $\mu$LOs have improved generalization to depth and training horizon.

- **Authors’ reply to oqd9:** We point the reviewer to our experiments in section F.1.1 where we elaborate further on possible causes of the observed improvements in generalization to depth and training horizon.

&nbsp;
---
**References**
---
[1][Don't be lazy: CompleteP enables compute-efficient deep transformers, NeurIPS 2025]


[2][scaling exponents across parameterizations and optimizers, ICML 2024]

---

### Meta-Review · Area_Chair_aiNS · 2025-12-22

**Summary:**

This paper introduces $\mu$LO, a method that applies Maximal Update Parametrization ($\mu$P) to Learned Optimizers (LOs) to address the challenge of meta-generalization, particularly when applying LOs to neural networks that are wider than those encountered during meta-training. The authors derive the specific $\mu$P formulations for two state-of-the-art LO architectures (VeLO and small_fc_lopt) and propose a meta-training recipe that enables zero-shot generalization to wider unseen tasks. Empirical evaluations demonstrate that $\mu$LOs significantly outperform standard parameterization baselines on wide networks and exhibit surprising generalization capabilities to deeper networks (5x meta-training depth) and longer training horizons (25x meta-training length).

The reviewers initially expressed mixed reactions (scores of 2, 6, 6, 6), with Reviewer tB8g recommending rejection due to concerns about "meta-overfitting" and limited novelty. However, the rebuttal phase was highly effective; the authors provided new experimental data (test loss tables, ResNet experiments) and clarifications that resolved the primary concerns. Reviewer tB8g explicitly agreed to raise their score, and the authors addressed the specific requests of the other reviewers, positioning the paper for acceptance.

Based on the successful rebuttal phase where the most critical reviewer explicitly agreed to raise their score and all requested ablation studies (specifically on ResNets) were provided, the AC recommends Acceptance.

**Reviewer Concerns:**

**Resolved Concerns**
- Meta-Overfitting (Reviewer tB8g): The reviewer initially interpreted Figure 5 as evidence of meta-overfitting (good performance on meta-train tasks, poor on unseen). The authors clarified that the figure actually demonstrates superior generalization compared to baselines that diverge.
- Lack of Test Metrics (Reviewer tB8g): The reviewer criticized the reliance on training curves. The authors addressed this by providing new tables for ImageNet ViT and LM1B, showing that final test loss closely tracks training loss, confirming that the optimizees were not overfitting.
- Limited Novelty (Reviewer tB8g): The reviewer felt the method was a direct application of prior $\mu$P work. The authors successfully argued that deriving $\mu$P for specific LO architectures is novel and established a new connection between LO meta-generalization and hyperparameter transfer.
- Lack of CNN Experiments (Reviewer P3eP): The reviewer requested validation on Convolutional Neural Networks. The authors added Section F.1.2 to the appendix, presenting results on ResNet tasks where $\mu$LO outperformed the baseline4444.
- Missing References (Reviewer PtQP): The reviewer noted missing citations regarding "CompleteP" and scaling exponents. The authors incorporated these into the Related Work and Future Work sections.

**Outstanding Concerns**

- Theoretical Basis for Depth/Horizon Generalization (Reviewer oqd9): The reviewer noted that the explanation for the method's success on deeper networks and longer horizons was speculative (relying on pre-activation stability analysis) rather than theoretical. The authors acknowledged this as an empirical finding that lacks a formal theoretical proof in the current work.
- Meta-Training Diversity (Reviewer oqd9): The reviewer asked about scaling meta-training to diverse architectures. The authors acknowledged this as a future direction but did not include such experiments due to computational costs.

**Reviewer Scores:**

- Reviewer tB8g (2 to 6). The reviewer explicitly stated, "all my concerns are addressed. I will raise my score." Given the shift from "Reject," a move to "Weak Accept" is the most logical outcome.
- Reviewer PtQP (6 to 7). The reviewer already rated Soundness and Presentation as excellent. With their minor requests (references and compute context) fully addressed, a score increase to "Accept" is expected.
- Reviewer P3eP (6 to 6). This reviewer had a single substantial request (add CNN experiments). Since the authors added a dedicated section and table for ResNet results, the reviewer's condition for a higher score was met.
- Reviewer oqd9 (6 to 6). While the authors responded to the questions, the primary critique regarding the "speculative" nature of the depth generalization results is an inherent limitation of the empirical study. The score is likely to remain stable.

---

### Decision · Program_Chairs · 2026-01-26

Accept (Poster)